# Physics Informed Distillation for Diffusion Models

**Joshua Tian Jin Tee\* & Kang Zhang\***  *{joshuateetj,zhangkang}@kaist.ac.kr*
*School of Electrical Engineering*
*Korea Advanced Institute of Science and Technology (KAIST)*

**Hee Suk Yoon**  *hskyoon@kaist.ac.kr*
*School of Electrical Engineering*
*Korea Advanced Institute of Science and Technology (KAIST)*

**Dhananjaya Nagaraja Gowda**  *d.gowda@samsung.com*
*Samsung Research*

**Chanwoo Kim**  *chanwcom@korea.ac.kr*
*Department of Artificial Intelligence*
*Korea University*

**Chang D. Yoo**[†]  *cd_yoo@kaist.ac.kr*
*School of Electrical Engineering*
*Korea Advanced Institute of Science and Technology (KAIST)*

**Reviewed on OpenReview:** *https://openreview.net/forum?id=rOvaUsF996*

## Abstract

Diffusion models have recently emerged as a potent tool in generative modeling. However, their inherent iterative nature often results in sluggish image generation due to the requirement for multiple model evaluations. Recent progress has unveiled the intrinsic link between diffusion models and Probability Flow Ordinary Differential Equations (ODEs), thus enabling us to conceptualize diffusion models as ODE systems. Simultaneously, Physics Informed Neural Networks (PINNs) have substantiated their effectiveness in solving intricate differential equations through implicit modeling of their solutions. Building upon these foundational insights, we introduce Physics Informed Distillation (PID), which employs a student model to represent the solution of the ODE system corresponding to the teacher diffusion model, akin to the principles employed in PINNs. Through experiments on CIFAR 10 and ImageNet 64x64, we observe that PID achieves performance comparable to recent distillation methods. Notably, it demonstrates predictable trends concerning method-specific hyperparameters and eliminates the need for synthetic dataset generation during the distillation process. Both of which contribute to its easy-to-use nature as a distillation approach for Diffusion Models. Our code and pre-trained checkpoint are publicly available at: `https://github.com/pantheon5100/pid_diffusion.git`.

## 1 Introduction

Diffusion models (Sohl-Dickstein et al., 2015; Song et al., 2021b; Ho et al., 2020) have demonstrated remarkable performance in various tasks, including image synthesis (Dhariwal & Nichol, 2021; Nichol et al., 2022; Ramesh et al., 2022; Saharia et al., 2022a), semantic segmentation (Baranchuk et al., 2022; Wolleb et al., 2022; Kirillov et al., 2023), and image restoration (Saharia et al., 2022b; Whang et al., 2022; Li et al., 2022; Niu et al., 2023). With a more stable training process, it has achieved better generation results

---

\* These authors contributed equally to this work and are listed in alphabetical order.
† Corresponding Author.

that outperform other generative models, such as GAN (Goodfellow et al., 2020), VAE (Kingma & Welling, 2013), and normalizing flows (Kingma & Dhariwal, 2018). The success of diffusion models can mainly be attributed to their iterative sampling process which progressively removes noise from a randomly sampled Gaussian noise. However, this iterative refinement process comes with the huge drawback of low sampling speed, which strongly limits its real-time applications (Salimans & Ho, 2022; Song et al., 2023).

Recently, Song et al. (2021b) and Karras et al. (2022) have proposed viewing diffusion models from a continuous time perspective. In this view, the forward process that takes the distribution of images to the Gaussian distribution can be viewed as a stochastic differential equation (SDE). On the other hand, diffusion models learn the associated backward SDE through score matching. Interestingly, Song et al. (2021b) demonstrate that diffusion models can also be used to model a probability flow ODE system that is equivalent in distribution to the marginal distributions of the SDE. In addition, Physics Informed Neural Networks (PINNs) have proven effective in solving complex differential equations (Raissi et al., 2019; Cuomo et al., 2022) by learning the underlying dynamics and relationships encoded in the equations.

Building upon these developments, we propose a distillation method for diffusion models called Physics Informed Distillation (PID), a method that takes a PINNs-like approach to distill a single-step diffusion model. Our method trains a model to predict the trajectory at any point in time given the initial condition relying solely on the ODE system. During training, we view the teacher diffusion model as an ODE system to be solved by the student model in a physics-informed fashion. In this framework, the student model approximates the ODE trajectories, as illustrated in Figure 1, without explicitly observing the images in the trajectory. In detail, our contributions can be summarized as follows:

- We propose and analyze Physics Informed Distillation (PID), a knowledge distillation technique heavily inspired by PINNs that enables single-step image generation, providing theoretical bounds for the method.

- Through experiments on CIFAR-10 and ImageNet 64x64, we showcase our approaches' effectiveness in generating high-quality images with only a single forward pass.

- We demonstrate that similar to PINNs where the performance improvements saturate at a sufficiently large number of collocation points, our approach with a high enough discretization number performs best, showcasing its potential as a knowledge distillation approach that does not need additional tuning of method specific hyperparameters.

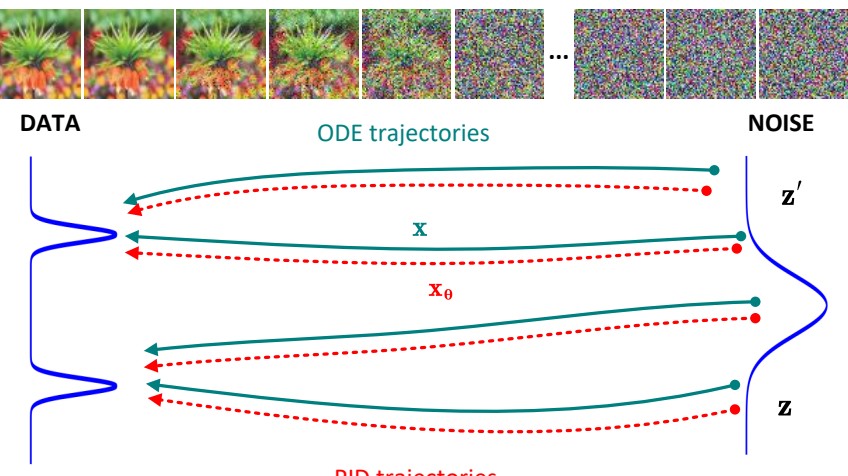

Figure 1: An overview of the proposed method, which involves training a model $\mathbf{x}_\theta(\mathbf{z}, \cdot)$ to approximate the true trajectory $\mathbf{x}(\mathbf{z}, \cdot)$.

## 2 Related Works

The remarkable performance of diffusion models in various image generation tasks has garnered considerable attention. Nevertheless, the slow sampling speed associated with these models remains a significant drawback. Consequently, researchers have pursued two primary avenues of exploration to tackle this challenge: training-free and training-based methods.

**Training-free methods**. Several training-free methods have been proposed in the field of diffusion model research to expedite the generation of high-quality images. Notably, Lu et al. (2022) introduced DPM-Solver, a high-order solver for diffusion ODEs. This method significantly accelerates the sampling process of diffusion probabilistic models by analytically computing the linear part of the solution. With only ten steps, DPM-Solver achieves decent performance compared to the usual requirement of hundreds to thousands of function evaluations. Another noteworthy approach is DEIS (Zhang & Chen, 2023), which exploits the semilinear structure of the empirical probability flow ODE, as in DPM-Solver, to enhance sampling efficiency. These methods demonstrate the surprising ability to significantly reduce the number of model evaluations while remaining training-free. On another note, Karras et al. (2022) propose enhancements to the SDE and probability flow ODE systems by implementing higher order solvers. Jolicoeur-Martineau et al. (2021) also presents higher order solvers for SDE systems, aimed at diminishing the number of functional evaluations required in a training free fashion. However, it is important to note that they only mitigate the core issue of the iterative sampling nature of diffusion models without eliminating it entirely. In contrast to these training-free methods, training-based approaches have been proposed to perform single-step inference while aiming to maintain the performance of the teacher diffusion model. Our paper focuses on this particular field of study.

**Training-based methods**. In the domain of diffusion model distillation, training-based strategies can be broadly classified into two primary groups: those relying on synthetic data and those trained with original data. Among the former, noteworthy methods such as Knowledge Distillation (Luhman & Luhman, 2021), DSNO (Zheng et al., 2023), and Rectified Flow (Liu et al., 2023) have recently exhibited their efficacy in distilling single-step student models. However, a notable drawback inherent in these approaches is the computationally costly nature of generating such synthetic data, which presents scalability challenges, particularly for larger models. In response, methods using only the original dataset such as Salimans & Ho (2022) (Progressive Distillation) and Song et al. (2023) (Consistency Models) have emerged as solutions to this issue. Progressive Distillation adopts a multi-stage distillation approach, while Consistency Models employ a self-teacher model, reminiscent of self-supervised learning techniques.

Another notable mention is the recent data-free distillation approach BOOT (Gu et al., 2023), which solely relies on a teacher model without requiring synthetic data, similar to our proposed approach. However, there are fundamental differences between our work and BOOT. While our focus lies on solving the original probability flow ODE system, BOOT concentrates on Signal ODEs. In tackling the original ODE, we address challenges such as Lipschitz explosion problems near the origin, which can cause instability during training. We mitigate these issues through our parametrization method outlined in Appendix A.4. On the contrary, BOOT's focus on Signal ODEs alleviates this particular challenge but introduces a different issue: the inability to satisfy boundary conditions through hard constraints. Consequently, starting from the distinct ODE systems addressed by BOOT (Gu et al., 2023) and our approach, our methods diverge due to the unique difficulties associated with each ODE system..

Another relevant work is the recent CTM (Kim et al., 2024). It's important to note that CTM shares a similar perspective with CT (Song et al., 2023), focusing on distilling the paths traced by numerical solvers. On the other hand, we approach training from a PINNs (Physics-Informed Neural Networks) perspective, aiming to minimize the residual loss associated with the ODE system during training. It's worth highlighting that the similarities between CTM and PID are primarily observed in the Euler method due to its reliance on first-order gradient approximation. However, when considering second-order numerical gradient approximations, as presented in Table 3, our method demonstrates effectiveness despite its departure from CTM's training paradigm due to its basis in the PINN paradigm. Consequently, CTM (Kim et al., 2024) and PID can be seen as methods advocating from distinct perspectives, with their equivalence being tangential primarily in the first-order setting.

## 3 Preliminaries

### 3.1 Diffusion Models

Physics Informed Knowledge Distillation is heavily based on the theory of continuous time diffusion models by Song et al. (2021b). These models describe the dynamics of a diffusion process through a stochastic differential equation:

$$\mathrm{d}\mathbf{x} = \mathbf{f}(\mathbf{x}, t)\mathrm{d}t + g(t)\mathrm{d}\boldsymbol{w}_t, \tag{1}$$

where $t \in [0, T]$, $\boldsymbol{w}_t$ is the standard Brownian motion (Uhlenbeck & Ornstein, 1930; Wang & Uhlenbeck, 1945) (a.k.a Wiener process), $\mathbf{f}(\cdot, \cdot)$ and $g(\cdot)$ denote the drift and diffusion coefficients, respectively. The distribution of $\mathbf{x}$ is denoted as $p_t(\mathbf{x})$ with the initial distribution $p_0(\mathbf{x})$ corresponding to the data distribution, $p_{\mathrm{data}}$.

As proven in Song et al. (2021b), this diffusion process has a corresponding probability flow ODE with the same marginal distributions $p_t(\mathbf{x})$ of the form:

$$\mathrm{d}\mathbf{x} = \left[\mathbf{f}(\mathbf{x}, t) - \frac{1}{2}g(t)^2 \nabla_{\mathbf{x}} \log p_t(\mathbf{x})\right] \mathrm{d}t, \tag{2}$$

where $\nabla_{\mathbf{x}} \log p_t(\mathbf{x})$ denotes the score function. Diffusion models learn to generate images through score matching (Song et al., 2021b), approximating the score function as $\nabla_{\mathbf{x}} \log p_t(\mathbf{x}) \approx \mathbf{s}_\phi(\mathbf{x}, t)$ with $\mathbf{s}_\phi(\mathbf{x}, t)$ being the score model parameterized by $\phi$. As such, diffusion models learn to model the probability flow ODE system of the data while relying on iterative finite solvers to approximate the modeled ODE.

### 3.2 Physics Informed Neural Networks (PINNs)

PINNs are a scientific machine-learning technique that can solve any arbitrary known differential equation (Raissi et al., 2019; Cuomo et al., 2022). They rely heavily on the universal approximation theorem (Hornik et al., 1989) of neural networks to model the solution of the known differential equation (Cuomo et al., 2022). To better explain the learning scheme of PINNs, let us first consider an ODE system of the form:

$$\frac{\mathrm{d}\mathbf{x}}{\mathrm{d}t} = u(\mathbf{x}, t),$$
$$\mathbf{x}(T) = x_0, \tag{3}$$

where $t \in [0, T]$, $u(\cdot, \cdot)$ is an arbitrary continuous function and $x_0$ is an arbitrary initial condition at the time $T$. To solve this ODE system, Physics Informed Neural Networks (PINNs) can be divided into two distinct approaches: a soft conditioning approach (Cuomo et al., 2022), in which boundary conditions are sampled and trained, and a hard conditioning approach, (Lagaris et al., 1998; Cuomo et al., 2022) where such conditions are automatically satisfied through the utilization of skip connections. For the sake of simplicity, we exclusively focus on the latter easier approach, where the PINNs output is represented as follows:

$$\mathbf{x}_\theta(t) = c_{\mathrm{skip}}(t)x_0 + c_{\mathrm{out}}(t)\mathbf{X}_\theta(t),$$
$$\text{where } c_{\mathrm{skip}}(T) = 1 \text{ and } c_{\mathrm{out}}(T) = 0, \tag{4}$$

here $\mathbf{X}_\theta$ denotes a neural network parametrized by $\theta$ and the functions, $c_{\mathrm{skip}}$ and $c_{\mathrm{out}}$, are chosen such that the boundary condition is always satisfied. Following this, PINNs learn by reducing the residual loss denoted as:

$$\mathcal{L} = \left\| \frac{\mathrm{d}\mathbf{x}_\theta(t)}{\mathrm{d}t} - u(\mathbf{x}_\theta(t), t) \right\|^2. \tag{5}$$

Through this, they can model physical phenomena represented by such ODE systems. Inspired by PINNs ability to solve complex ODE systems (Lagaris et al., 1998; Cuomo et al., 2022), we use a PINNs-like approach to perform Physics Informed Distillation. This distillation approach uses the residual loss in PINNs to solve the probability flow ODE system modeled by diffusion models. Through this distillation, the student trajectory function in Figure 1 can perform fast single-step inference by querying the end points of the trajectory.

## 4 Physics Informed Knowledge Distillation

### 4.1 Trajectory Functions

In this section, we begin by introducing the concept of trajectory functions, representing solutions to a given arbitrary ODE system. Subsequently, we demonstrate how these trajectory functions can be learned through the application of a simple PINNs loss, resulting in a single-step sampler. It's noteworthy that a recent work, BOOT, also employs a PINNs-like approach to distill trajectory functions. However, their primary focus is on distilling the Signal ODE, while our emphasis lies in the direct distillation of the probability flow ODE. Later sections will delve into specific modifications, highlighting challenges encountered and successful adaptations made to effectively distill the probability flow ODE.

From this point onwards, we adopt the configuration introduced in EDM (Karras et al., 2022). The configuration utilizes the empirical probability flow ODE on the interval $t \in [\epsilon, T]$ of the following form:

$$
\begin{aligned}
\frac{\mathrm{d}\mathbf{x}}{\mathrm{d}t} &= -t\mathbf{s}_\phi(\mathbf{x}, t) \\
&= \frac{\mathbf{x} - D_\phi(\mathbf{x}, t)}{t},
\end{aligned}
\tag{6}
$$

where $\mathbf{x}(T) \sim \mathcal{N}(\mathbf{0}, T^2\mathbf{I})$ , $\epsilon = 0.002$, $T = 80$ and $D_\phi(\mathbf{x}, t)$ is the teacher diffusion model. Under the assumption that Diffusion Models are Lipschitz continuous on the given time interval, this ODE system exhibits a unique solution (Grant, 1999). As such, we can conceptualize each noise as having its uniquely associated trajectory function, as illustrated in Figure 1. We denote this trajectory function as $\mathbf{x}(\mathbf{z}, t)$, where $\mathbf{z}$ represents the noise's initial condition at time $T$. Our objective is to train a model $\mathbf{x}_\theta(\mathbf{z}, t)$ that precisely captures the actual trajectories $\mathbf{x}(\mathbf{z}, t)$ for any arbitrary noise, $\mathbf{z}$. To achieve this, we adopt a PINNs-like approach to learn the trajectory functions $\mathbf{x}(\mathbf{z}, t)$. Specifically, we can minimize the residual loss, similar to the methodology employed in PINNs.

Similar to in PINNs, we require our solution function, $\mathbf{x}_\theta(\cdot, \cdot)$, to satisfy the boundary condition $\mathbf{x}_\theta(\mathbf{z}, T) = \mathbf{z}$. In PINNs literature, the two common approaches to achieve this is either by using a soft condition (Cuomo et al., 2022), in which boundary conditions are sampled and trained or through a strict condition (Lagaris et al., 1998) where these conditions are arbitrarily satisfied using skip connections. Since the use of skip connections is also common in diffusion training (Karras et al., 2022), we take inspiration from both fields and parametrize our model as:

$$
\begin{aligned}
\mathbf{x}_\theta(\mathbf{z}, t) &= c_{\text{skip}}(t)\,\mathbf{z} + c_{\text{out}}(t)\,\mathbf{X}_\theta\left(c_{\text{in}}(T)\,\mathbf{z}, c_{\text{noise}}(t)\right), \\
\text{where } c_{\text{skip}}(t) &= \frac{t}{T}, c_{\text{out}}(t) = \frac{T-t}{T}, c_{\text{in}}(T) = \frac{1}{\sqrt{0.5^2 + T^2}} \text{ and } c_{\text{noise}}(t) = \frac{\ln t}{4}.
\end{aligned}
\tag{7}
$$

Here, $\mathbf{X}_\theta$ denotes the neural network to be trained(insight into the choice of skip connection functions are provided in Appendix A.4.) Using the vanilla PINNs loss on the probability flow ODE, the loss is given as:

$$
\mathcal{L}_{\text{PINNs}} = \mathbb{E}_{i,\mathbf{z}}\left[d\left(\frac{\mathrm{d}\mathbf{x}_\theta(\mathbf{z}, t_i)}{\mathrm{d}t_i}, \frac{\mathbf{x}_\theta(\mathbf{z}, t_i) - D_\phi\left(\mathbf{x}_\theta(\mathbf{z}, t_i), t_i\right)}{t_i}\right)\right]
\tag{8}
$$

where $d(\cdot, \cdot)$ is any arbitrary distance metric (e.g. L2 and LPIPS). In diffusion models, the Lipschitz constant of the ODE systems they model tend to explode to infinity near the origin (Yang et al., 2023). For instance, in Equation 6, it can be seen that the ODE system explodes as $t$ approaches 0. Consequently, training a solver in a vanilla physics informed fashion may yield suboptimal performance, as it involves training the gradients of our student model to correspond with an exploding value. To alleviate this, we shift the variables as:

$$
\mathcal{L}_{\text{PINNs}} = \mathbb{E}_{i,\mathbf{z}}\left[d\left(\mathbf{x}_\theta(\mathbf{z}, t_i) - t_i\frac{\mathrm{d}\mathbf{x}_\theta(\mathbf{z}, t_i)}{\mathrm{d}t_i}, D_\phi\left(\mathbf{x}_\theta(\mathbf{z}, t_i), t_i\right)\right)\right].
\tag{9}
$$

Through this, we can obtain a stable training loss that does not explode when time values near the origin are sampled. By performing this straightforward variable manipulation, we can interpret the residual loss as

the process of learning to match the composite student model, parametrized by $\theta$, on the left portion of the distance metric with the output of the teacher diffusion model on the right portion of the distance metric. Since the output of the teacher diffusion model on the right is also dependent on the student model, we can also view this from a self-supervised perspective. In this perspective, it is often conventional to stop the gradients flowing from the teacher model (Grill et al., 2020; Caron et al., 2021; Chen et al., 2020). In line with this, we also stop the gradients flowing from the teacher diffusion model during training.

## 4.2   Numerical Differentiation

In what follows, we will briefly overview the challenges associated with automatic differentiation for gradient computation. Then, we propose the adoption of numerical differentiation, a technique prevalent in standard PINNs training and also employed in the BOOT framework.

The residual loss in PINNs requires computing gradients of the trajectory function with respect to its inputs, which can be computationally expensive using forward-mode backpropagation and may not be readily available in certain packages. Furthermore, studies such as Chiu et al. (2022) have demonstrated that training PINNs using automatic differentiation can lead to convergence to unphysical solutions. To address these challenges, we employ a different approach by relying on a straightforward numerical differentiation method to approximate the gradients. Specifically, we utilize a first-order upwind numerical approximation, given as:

$$\frac{\mathrm{d}\mathbf{x}_\theta(\mathbf{z}, t)}{\mathrm{d}t}\bigg|_{\text{num}} = \frac{\mathbf{x}_\theta(\mathbf{z}, t) - \mathbf{x}_\theta(\mathbf{z}, t - \Delta t)}{\Delta t} \ . \tag{10}$$

By employing this numerical differentiation scheme, we can efficiently estimate the gradients needed for the training process. This approach offers a simpler alternative to costly forward-mode backpropagation and mitigates the issues related to convergence to unphysical solutions observed in automatic differentiation-based training of PINNs.

---

**Algorithm 1** Physics Informed Distillation Training

---

**Input:** Trained teacher model $D_\phi$, PID model $\mathbf{x}_\theta$, LPIPS loss $d(\cdot, \cdot)$, learning rate $\eta$, discretization number $N$.

1: $\theta \leftarrow \phi$ // Initialize student from teacher
2: **repeat**
3:   $i \sim U[0, 1, ..., N]$ // Sample time index i.i.d from a uniform distribution
4:   $\mathbf{z} \sim \mathcal{N}(\mathbf{0}, T^2\mathbf{I})$ // Sample data
5:   $\frac{\mathrm{d}\mathbf{x}}{\mathrm{d}t} \leftarrow (\mathbf{x}_\theta(\mathbf{z}, t_i) - \mathbf{x}_\theta(\mathbf{z}, t_{i+1}))/(t_i - t_{i+1})$ // Numerical gradient approximation
6:   $\mathbf{x}_{\text{teacher}} \leftarrow \text{sg}\left(D_\phi\left(\mathbf{x}_\theta(\mathbf{z}, t_i), t_i\right)\right)$ // Get teacher model output
7:   $\mathcal{L}_{\text{PID}} \leftarrow d\left(\mathbf{x}_{\text{teacher}}, \mathbf{x}_\theta(\mathbf{z}, t_i) - t_i * \frac{\mathrm{d}\mathbf{x}}{\mathrm{d}t}\right)$ // Loss calculation according to Equation 11
8:   $\theta \leftarrow \theta - \eta\nabla_\theta\mathcal{L}_{\text{PID}}$ // Update weights
9: **until** model converged

---

## 4.3   Physics Informed Distillation

In this section, we consolidate the insights gathered from the preceding sections to present Physics Informed Distillation (PID) as a distillation approach for single-step sampling. In addition, we conduct a theoretical analysis of our proposed PID loss, establishing bounds on the generation error of the distilled PID student model.

Replacing the exact gradients with the proposed numerical differentiation and setting $\Delta t = t_i - t_{i+1}$, we obtain the Physics Informed Distillation loss:

$$\mathcal{L}_{\text{PID}} = \mathbb{E}_{i,\mathbf{z}}\left[d\left(\mathbf{x}_\theta(\mathbf{z}, t_i) - t_i\frac{\mathbf{x}_\theta(\mathbf{z}, t_i) - \mathbf{x}_\theta(\mathbf{z}, t_{i+1})}{t_i - t_{i+1}}, \text{sg}\left(D_\phi(\mathbf{x}_\theta(\mathbf{z}, t_i), t_i)\right)\right)\right] \tag{11}$$

Table 1: FID and IS table comparisons for various sampler based and distillation based methods on CIFAR-10. The asterisk (*) denotes methods that require the generation of synthetic dataset. The neural function evaluations (NFE), FID score and IS values reported where obtained from the respective papers.

| METHOD | Distance Metric | NFE ($\downarrow$) | FID ($\downarrow$) | IS ($\uparrow$) |
|---|---|---|---|---|
| EDM (Karras et al., 2022) | | 36 | 2.04 | |
| EDM+Euler Solver (Karras et al., 2022) | | 250 | 2.10 | |
| DDPM Ho et al. (2020) | | 1000 | 3.17 | 9.46 |
| DDIM (Song et al., 2021a) | | 10 | 13.36 | |
| | | 50 | 4.67 | |
| DPM-Solver-Fast (Lu et al., 2022) | | 10 | 4.70 | |
| 3-DEIS (Zhang & Chen, 2023) | | 10 | 4.17 | |
| Teacher DDPM | | | | |
| Progressive Distillation (Salimans & Ho, 2022) | L2 | 1 | 8.34 | 8.69 |
| DSNO* (Zheng et al., 2023) | LPIPS | 1 | 3.78 | - |
| Teacher Rectified Flow | | | | |
| 2-Rectified Flow (+ distill)* (Liu et al., 2023) | LPIPS | 1 | 4.85 | 9.01 |
| Teacher EDM | | | | |
| Consistency Model (Song et al., 2023) | LPIPS | 1 | 3.55 | 9.48 |
| BOOT (Gu et al., 2023) | LPIPS | 1 | 4.38 | - |
| Diff-Instruct (Luo et al., 2023) | L2 | 1 | 4.12 | 9.89 |
| Equilibrium Models (Geng et al., 2023) | L1 | 1 | 6.91 | 9.16 |
| **PID** (Ours) | LPIPS | 1 | **3.92** | **9.13** |

where sg($\cdot$) denotes the stop gradient operation. For the time-space discretization scheme, we follow the same scheme as in EDM (Karras et al., 2022). The discretization error added in such a scheme is bounded as shown in the Lemma 1, and we provide the proof in Appendix A.2.

**Lemma 1.** *Assuming $D_\phi(\mathbf{x}, t)$ is Lipchitz continuous with respect to $\mathbf{x}$, if $\mathcal{L}_{PID} = 0$, $||\mathbf{x}_\theta(\mathbf{z}, t) - \mathbf{x}(\mathbf{z}, t)||_2 \leq \mathcal{O}(\Delta t)$, where $\Delta t = \max_{i \in [0, N-1]} |t_{i+1} - t_i|$.*

An intriguing aspect of this theorem lies in its connection to Euler solvers and first-order numerical gradient approximations. Specifically, when our model achieves a loss of 0, the approximate trajectories traced by the PID-trained model, denoted as $\mathbf{x}_\theta(\mathbf{z}, t)$, effectively mirrors those obtained using a simple Euler solver employing an equivalent number of steps, denoted as $N$. Moreover, as indicated in Lemma 1, the discretization error is well controlled and diminishes with an increasing number of discretization steps, $N$. This observation suggests that by employing a higher number of discretization steps, we can effectively minimize the error associated with the discretization process, achieving improved performance.

In practice, we replace the traditional distance metric with LPIPS, motivated by the successes of Rectified Flow Distillation (Liu et al., 2023) and Consistency Models (Song et al., 2023) using this metric. Despite not being a proper distance metric, LPIPS empirically produces the best results as it is less sensitive to pixel differences, allowing the model to focus its capacity on generating high-fidelity images without wasting capacity on imperceptible differences.

By employing the PID training scheme, the resulting model can effectively match the trajectory function $\mathbf{x}(\mathbf{z}, t)$ within the specified error bound mentioned earlier. Single-step inference can be performed by simply querying the value of the approximate trajectory function $\mathbf{x}_\theta(\mathbf{z}, t)$ at the endpoint $\epsilon$, $\mathbf{x}_\theta(\mathbf{z}, \epsilon)$. With this, we observe that by treating the teacher diffusion model as an ODE system, we can theoretically train a student model to learn the trajectory function up to a certain discretization error bound without data and perform fast single-step inference. Algorithm 1 provides the pseudo-code describing our Physics Informed Distillation training process. After training, single-step generation can be achieved by querying the learned trajectory functions at the origin, denoted as $\mathbf{x}_\theta(\mathbf{z}, t_{min})$.

Table 2: FID table comparisons for various distillation based methods on ImageNet 64x64. The asterisk (*) denotes methods that require the generation of a synthetic dataset.

| Method | Distance Metric | NFE ($\downarrow$) | FID ($\downarrow$) |
|---|---|---|---|
| ADM (Dhariwal & Nichol, 2021) | | 250 | 2.07 |
| EDM (Karras et al., 2022) | | 79 | 2.44 |
| EDM+Euler Solver (Karras et al., 2022) | | 250 | 2.41 |
| BigGAN-deep (Brock et al., 2019) | | 1 | 4.06 |
| Teacher DDPM | | | |
| Progressive Distillation (Salimans & Ho, 2022) | L2 | 1 | 15.39 |
| DSNO* (Zheng et al., 2023) | L1 | 1 | 7.83 |
| Teacher EDM | | | |
| Consistency Model (Song et al., 2023) | LPIPS | 1 | 6.20 |
| BOOT (Gu et al., 2023) | LPIPS | 1 | 12.3 |
| Diff-Instruct (Luo et al., 2023) | L2 | 1 | 4.24 |
| **PID** (Ours) | LPIPS | 1 | **9.49** |

## 5   Results

In this section, we empirically validate our theoretical findings through various experiments on CIFAR-10 (Krizhevsky & Hinton, 2009) and ImageNet 64x64 (Deng et al., 2009). The results are compared according to Frechet Inception Distance (FID) (Heusel et al., 2017b) and Inception Score (IS) (Salimans et al., 2016). All experiments for PID were initialized with the EDM teacher model, and all the competing methods were also initialized with their respective teacher diffusion model weights as noted in Table 1 and Table 2. In addition, unless stated otherwise, a discretization of 250 and LPIPS metric was used during training. More information on the training details can be seen in Appendix A.1.

We quantitatively compare the sample quality of our PID for diffusion models with other training-free and training-based methods for diffusion models, including DSNO (Zheng et al., 2023), Rectified Flow (Liu et al., 2023), PD (Salimans & Ho, 2022), CD (Song et al., 2023), BOOT (Gu et al., 2023) and Diff-Instruct (Luo et al., 2023). In addition to the baseline EDM (Karras et al., 2022) model, we make comparisons with other sampler-based fast generative models, such as DDIM (Song et al., 2021a), DPM-solver (Lu et al., 2022), and DEIS (Zhang & Chen, 2023). In Table 1 we show our results on CIFAR 10 dataset. In this, PID maintains a competitive result in both FID and IS with the most recent single-step generation methods, achieving an FID of 3.92 and IS of 9.13, while outperforming a few. In particular, we outperform Diff-Instruct, Rectified Flow, PD, Equilibrium Models and BOOT (Gu et al., 2023) by a decent margin on CIFAR-10. On the other hand, we maintain a competitive performance with DSNO and CD, only losing to it by a small margin.

Table 2 presents the results obtained from experiments conducted on ImageNet 64x64. Analyzing Table 2, we observe that our method surpasses PD (Salimans & Ho, 2022), achieving an FID of 9.49. Nevertheless, echoing our observations from the CIFAR-10 experiments, our approach lags behind DSNO (Zheng et al., 2023) and CD (Song et al., 2023), which achieves a lower FID of 7.83 and 6.20 respectively. Despite this, it is worth emphasizing that our method does not entail the additional costs associated with generating expensive synthetic data, which is a characteristic feature of DSNO. This cost-effective aspect represents a notable advantage of our approach in the context of ImageNet 64x64 experiments despite its poorer performance.

In Figure 2, we compare qualitatively between the images generated from the teacher EDM model and the student model. From this, we observe that in general, the student model aligns with the teacher model images, generating similar images for the same noise seed. Additional samples from Imagenet and CIFAR-10 are provided in Appendix A.6.

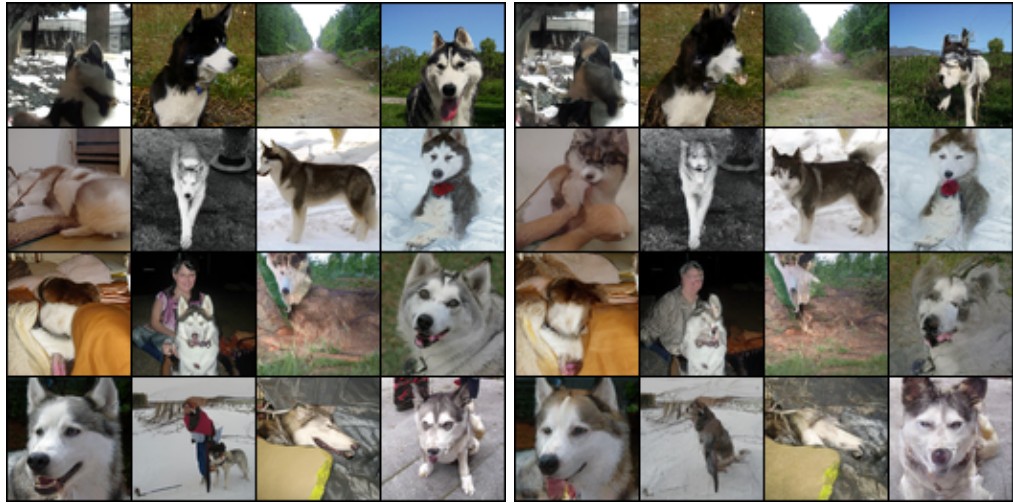

(a) Random samples from EDM             (b) Random samples from PID

Figure 2: Conditional image generation comparison on ImageNet 64×64 for the same seed with the same class label "Siberian husky". Left panel: random samples generated by teacher model EDM. Right panel: generated by student PID model.

Table 3: FID table comparisons for different order numerical differentiation on CIFAR 10.

| Method | FID ($\downarrow$) |
|---|---|
| PID (1st Order) | 3.92 |
| PID (2nd order - Central Difference) | 3.68 |

## 6 Ablation on PID Design Choices and its properties

### 6.1 Comparing Higher Order Numerical Differentiation

In this section, we investigate the impact of higher-order numerical differentiation on the distillation performance of PID. Specifically, we compare the outcomes of employing 1st order numerical differentiation with those obtained using the 2nd order Central Difference Method. While many higher-order approaches typically necessitate more than 2 model evaluations, leading to increased computational costs for numerical gradient computation, the Central Difference method, despite being a second-order numerical differentiation technique, only requires 2 model evaluations, maintaining the same computational cost as the 1st order approach. As indicated in Table 3, the 2nd order approach exhibits a slightly superior performance compared to 1st order numerical differentiation. This aligns with observations from standard PINNs training, where higher-order numerical differentiation tends to yield solutions that closely align with the actual ODE system.

### 6.2 Automatic Differentiation vs Numerical Differentiation

In this section, we empirically evaluate the importance of Numerical Differentiation in the PID training scheme. The experiment presented in Figure 3 was conducted on CIFAR 10, employing the same experimental settings as in the main results, albeit with the substitution of numerical differentiation for automatic differentiation. From Figure 3, it can be observed that automatic differentiation in PID training leads to convergence towards unrealistic images characterized by high image contrast, resulting in poor FID scores. This observation aligns with findings in Chiu et al. (2022) where in PINNs dealing with systems of differential equations, numerical differentiation often outperforms automatic differentiation, even in regions with a large number of collocation points. Intuitively, numerical differentiation can be conceptualized as utilizing local

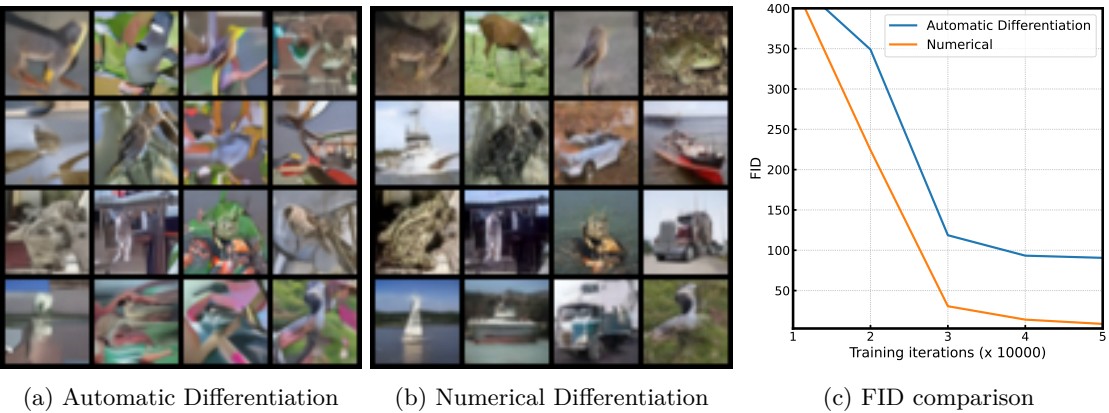

(a) Automatic Differentiation      (b) Numerical Differentiation      (c) FID comparison

Figure 3: Comparison between automatic differentiation and numerical differentiation on CIFAR-10 dataset.

points to stabilize the PINN loss, thereby preventing convergence to solutions too divergent from the ground truth ODE solutions.

## 6.3 Random Initialization

In this section, we examine the impact of random initialization on the performance of PID. The experiment detailed in Figure 4 was carried out using the same experimental settings as the main results on CIFAR 10, employing randomly initialized student weights. From Figure 4, it is evident that random initialization leads PID to converge to a suboptimal local point, resulting in a higher FID compared to initializations with pre-trained weights. Hence, validating the use of initializing the student model with the pre-trained teacher diffusion model instead of random initialization.

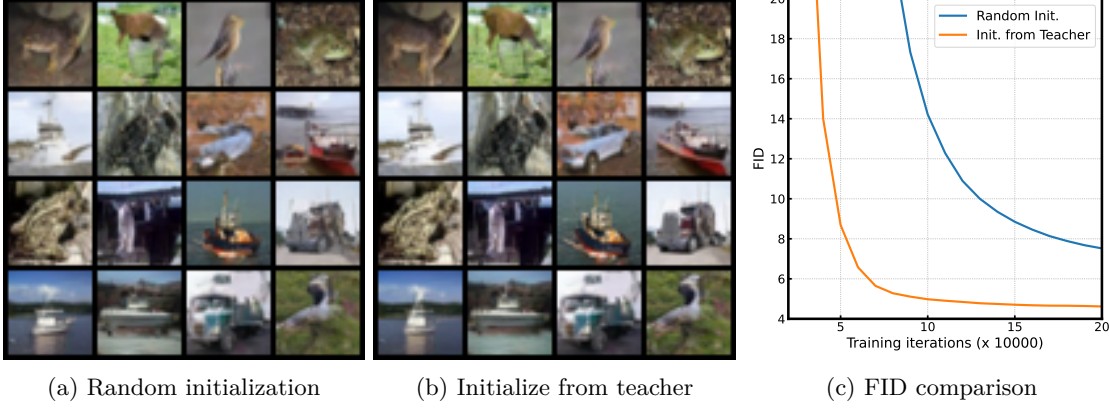

(a) Random initialization      (b) Initialize from teacher      (c) FID comparison

Figure 4: Comparison between student model weights random initialized and initialized with teacher model weights on CIFAR-10 dataset.

## 6.4 Stop Gradient

In this section, we investigate the impact of stop gradient in the PID training scheme. The experiment outlined in Figure 5 was carried out under the same experimental settings as the main results on CIFAR 10, albeit without the use of a stop gradient. From Figure 5, we observe a degradation in performance when backpropagating through the teacher diffusion model during PID training. We hypothesize that the removal of stop gradient which updates the student while backpropagation through the teacher diffusion

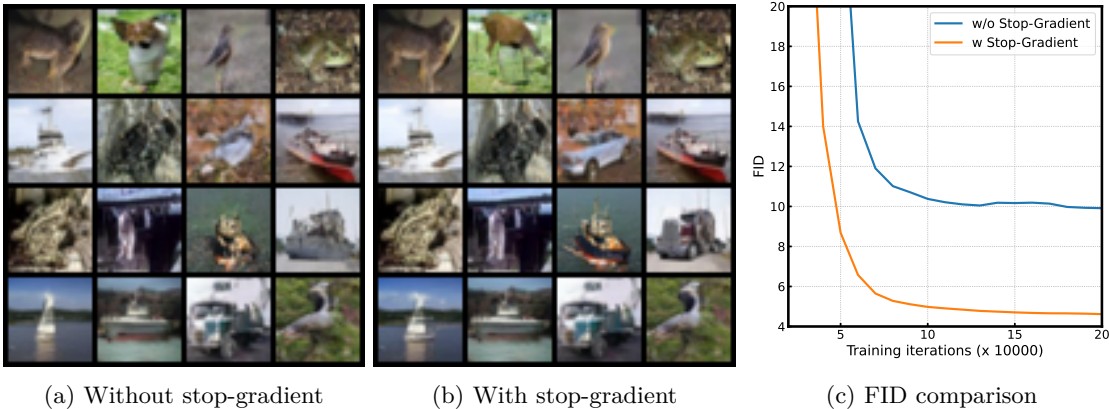

(a) Without stop-gradient      (b) With stop-gradient      (c) FID comparison

Figure 5: The impact of stop gradient in the PID training on CIFAR-10 dataset.

model behaves akin to an adversarial attack on the teacher model. This results in generated samples with low residual PINN loss but high levels of distortion in the images.

### 6.5 L2 vs LPIPS comparison

To thoroughly investigate the influence of different distance metrics on model performance, we conducted experiments on CIFAR-10 using the L2 and LPIPS metrics. These experiments were carried out following the same experimental setup described in the main results. Analyzing the results depicted in Figure 7, we observe a similar trend to previous studies utilizing LPIPS metric (Song et al., 2023; Liu et al., 2023), wherein a slower convergence rate with L2 compared to LPIPS was observed. This difference between L2 and LPIPS persists even up till convergence with the L2 metric obtaining an FID of 5.85 and the LPIPS metric achieving an FID of 3.92. Consequently, this reinforces the validity and suitability of the LPIPS metric in our training approach.

### 6.6 Comparing Discretization Numbers

To properly understand the effect of discretization number, $N$, on our method, the experiment on CIFAR-10 was repeated with different discretization values. The experiments conducted in this section were performed with the same hyperparameter setup as in the main results except with changing discretization number. The discretization values investigated here were $\{35, 80, 120, 200, 250, 500, 1000\}$.

In Figure 6, we can observe that the performance of our single step image generation model steadily increases with increasing discretization. This behaviour aligns well with the theoretical expectations according to Lemma 1 where the discretization error decreases with higher discretization. Additionally, this behaviour is also common to PINNs (Raissi et al., 2019) where increasing the collocation of points on a trajectory improves model performance. This stable and predictable trend with respect to discretization justifies our choice of setting our discretization number to 250 where the performance has plateaued, achieving good performance despite not tuning any methodology-specific hyperparameters.

Additionally, a higher discretization number has no effect on training time as seen in Figure 6 as it not only converges faster to a better FID value in the same number of iterations. Consequently, PID can be trained with an arbitrarily high discretization number to achieve optimal performance. This is in stark contrast to methods such as CD Song et al. (2023) where increasing or decreasing discretization numbers away from its optimal discretization number results in performance degradation resulting in the need to search this optimal discretization number that differs from dataset to dataset.

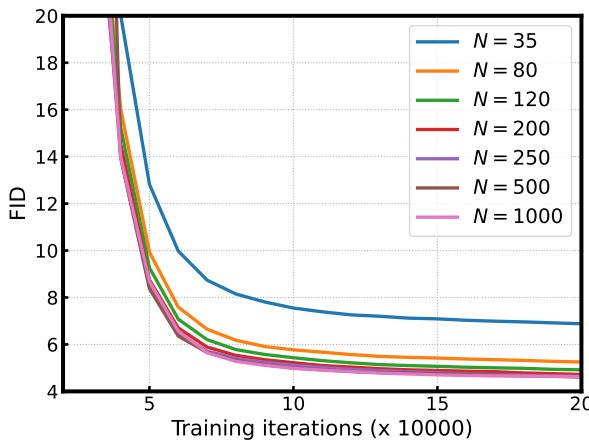
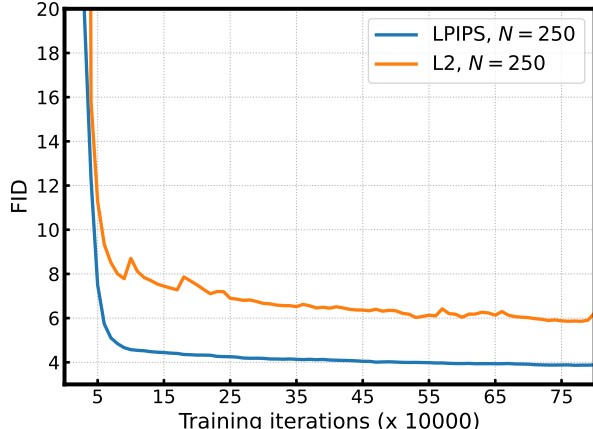

Figure 6: Training curve with different discretizations number on CIFAR-10.

Figure 7: Training curve for different distance metrics, L2 and LPIPS, on CIFAR-10.

## 7 Computational Costs comparisons

### 7.1 Training Efficiency Comparison

Due to the incorporation of numerical differentiation in our approach, we regrettably require two model evaluations, in contrast to the single model evaluations employed by other methods. In this section, we provide a detailed comparison of the training time efficiency of our method with recent works, CD (Song et al., 2023) and PD (Salimans & Ho, 2022). The analysis presented in Table 4 reveals an intriguing finding: despite the necessity of two functional evaluations, our approach incurs only a 35% higher training cost compared to doubling, as might be expected. This discrepancy arises from the fact that, while our method uses only a single teacher model evaluation, recent methods instead requires two iterative teacher model evaluations, thus incurring additional training expenses. It is worth emphasizing that despite the additional training cost per iteration attributed to our approach, the consistent and stable trend we observe concerning the discretization number allows us to train our model without the need to fine-tune any methodology-specific hyperparameters. This, in turn, significantly reduces the cost associated with hyperparameter optimization.

Table 4: Training time comparisons between PID and recent works on CIFAR 10.

| Method | Training Time (seconds per 10 iterations) |
| --- | --- |
| PD | 5.15 |
| CD | 5.21 |
| DSNO | 7.21 |
| **PID** (ours) | 7.13 |

### 7.2 Full Training Time comparisons with Recent Methods

In Table 5, we present a comparison of the full training time across several recent distillation methods for diffusion models. The comparisons were conducted on ImageNet 64x64, following the training settings outlined in the respective papers. As observed in the CIFAR-10 time comparisons in Table 4, our method is slower than CD due to the need for 2 functional evaluations for numerical differentiation.

Table 5: Training time comparison for each method, measured on 64 NVIDIA A100 GPUs. The asterisk(*) denotes the time taken to obtain the synthetic dataset used for distillation.

| Method | Training Iteration | Total Training Time (hours) |
|---|---|---|
| PD | 550000 | 104 |
| CM | 600000 | 144 |
| PID | 600000 | 197 |
| DSNO | 480000 | 96 + 24* |

## 8 Conclusion

In this paper, we introduce Physics Informed Distillation (PID), a method designed to train a single-step diffusion model that draws significant inspiration from Physics Informed Neural Networks (PINNs). Through a combination of empirical evaluations and theoretical underpinnings, we have demonstrated the robustness and competitiveness of our method in comparison to the majority of existing techniques. While it falls slightly behind DSNO and CD, it distinguishes itself by eschewing the need for costly synthetic data generation or meticulous tuning of methodology-specific hyperparameters. Instead, our approach achieves competitive performance with constant methodology-specific hyperparameters.

## 9 Acknowledgement

This work was supported by Institute of Information & communications Technology Planning & Evaluation (IITP) grant funded by the Korea government(MSIT) (No.2022-0-00184, Development and Study of AI Technologies to Inexpensively Conform to Evolving Policy on Ethics), and Institute for Information & communications Technology Planning & Evaluation (IITP) grant funded by the Korea government(MSIT) (No. 2021-0-01381, Development of Causal AI through Video Understanding and Reinforcement Learning, and Its Applications to Real Environments).

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

# A   Appendix

## A.1   Additional Experiment details

**Model Architectures** All pre-trained models utilized in our experiments were obtained from the EDM framework (Karras et al., 2022). Specifically, for the CIFAR-10 dataset, we employed the NCSN++ model architecture as described by (Song et al., 2021b). For experiments conducted on the ImageNet 64x64 dataset, we followed the architecture detailed by (Dhariwal & Nichol, 2021). For small student distillation experiment on CIFAR-10, the same architecture as the teacher model in (Song et al., 2021b) was used with the number of channels reduced from 128 to 64.

**Evaluation Metrics** For Frechet inception distance (FID, lower is better) (Heusel et al., 2017a) and Inception Score (IS, higher is better) (Salimans et al., 2016), 50,000 generated images were compared against their respective ground truth datasets. Three different seeds were employed, and the best result was selected since FID values typically exhibit approximately 2% variance between measurements (Karras et al., 2022).

**Training Details** For both CIFAR-10 and ImageNet, we use a pretrained EDM model (Karras et al., 2022) as our teacher model. Unless stated otherwise, all the student models $\mathbf{X}_\theta$ were initialized with the same weight as the pretrained EDM teacher model $D_\phi$. In the experiment involving the smaller student model, the student model was randomly initialized. Rectified Adam optimizer (Liu et al., 2020) was used for distillation with weight decay of 0 and a constant learning rate throughout the training iteration. Following (Karras et al., 2022), we use the EMA weights of student model for inference. The EMA decay value for both CIFAR-10 and ImageNet is same, 0.99995. Additional details on training hyperparameters are shown in Table 6. All the experiments are run with PyTorch (Paszke et al., 2019) on NVIDIA A100 GPU.

Unless explicitly stated otherwise, we utilized LPIPS as the default distance metric for training. When utilizing LPIPS as the distance metric $d(\cdot, \cdot)$, the input images were rescaled to 244x244 using bilinear upsampling before being fed into the VGG model (Simonyan & Zisserman, 2014).

Table 6: Hyperparameters used for the training runs.

| Hyperparameter | CIFAR-10 | ImagetNet 64x64 |
|---|---|---|
| Number of GPUs | 8xA100 | 32xA100 |
| Batch size | 512 | 2048 |
| Gradient clipping | - | ✓ |
| Mixed-precision (FP16) | - | ✓ |
| Learning rate $\times 10^{-4}$ | 2 | 1 |
| Dropout probability | 0% | 0% |
| EMA student model | 0.99995 | 0.99995 |

## A.2   Proof for Lemma 1

*Proof.* When $\mathcal{L}_{\mathrm{PID}}$ loss goes to 0, since the distance metric has the property,

$$x = y \Longleftrightarrow d(x, y) = 0. \tag{12}$$

We have the following equality $\forall \mathbf{z} \sim \mathcal{N}(\mathbf{0}, T^2 \mathbf{I}), \forall i \in [0, 1, ..., N-2]$,

$$\mathbf{x}_\theta(\mathbf{z}, t_i) - t_i \frac{\mathbf{x}_\theta(\mathbf{z}, t_i) - \mathbf{x}_\theta(\mathbf{z}, t_{i+1})}{t_i - t_{i+1}} = D_\phi(\mathbf{x}_\theta(\mathbf{z}, t_i), t_i)$$

$$\Delta t_i \mathbf{x}_\theta(\mathbf{z}, t_i) - t_i \mathbf{x}_\theta(\mathbf{z}, t_i) + t_i \mathbf{x}_\theta(\mathbf{z}, t_{i+1}) = \Delta t_i D_\phi(\mathbf{x}_\theta(\mathbf{z}, t_i), t_i) \text{ where } \Delta t_i = |t_{i+1} - t_i| \tag{13}$$

$$\mathbf{x}_\theta(\mathbf{z}, t_{i+1}) = \mathbf{x}_\theta(\mathbf{z}, t_i) - \Delta t_i \frac{-D_\phi(\mathbf{x}_\theta(\mathbf{z}, t_i), t_i)}{t_i}$$

Since this equality holds $\forall \mathbf{z} \sim \mathcal{N}(\mathbf{0}, T^2 \mathbf{I}), \forall i \in [0, 1, ..., N-2]$, the trajectory model $\mathbf{x}_\theta(\mathbf{z}, t_i)$ will have equivalent trajectories as that obtained through a euler solver. As such, it will have the same discretization

error bound, $\mathcal{O}(\Delta t)$.

$\square$

### A.3 PID with Model Compression

Table 7: FID table comparisons for small student model and large student model on CIFAR 10.

| Model | Parameters | FID ($\downarrow$) |
|---|---|---|
| Teacher (EDM) | 55.7M | 2.04 |
| Student (PID) | 55.7M | 3.93 |
| Student (PID) | 13.9M | 8.29 |

As is common in Knowledge Distillation literature, this field often involves model compression where a teacher model is used to produce a student model with comparable performance. In line with this paradigm, we train our model with the same hyperparameters as in the CIFAR-10 main results with less than a quarter of the model parameters. The small student architecture was constructed by reducing all the channels in the teacher models by half. More details on the small student architecture are provided in Appendix A.1. From Table 7, we can observe that despite the student model's significantly smaller size, it is still able to generate decent images, achieving an FID score of 8.29. Despite its drop in performance in contrast to the bigger student models, this result showcases the promising potential of Knowledge Distillation approaches in model compression and not just NFE reduction.

### A.4 Insight on Parametrization Choice for Physics Informed Distillation

To satisfy the boundary condition, we require a parametrization choice such that:

$$
\begin{aligned}
\mathbf{x}_\theta(\mathbf{z}, t) = & c_{\text{skip}}(t)\, \mathbf{z} + c_{\text{out}}(t)\, \mathbf{X}_\theta\left(c_{\text{in}}(T)\, \mathbf{z}, c_{\text{noise}}(t)\right), \\
& \text{where } c_{\text{out}}(T) = 0.
\end{aligned}
\tag{14}
$$

A straightforward selection for this would be a linear function, such as $T - t$. However, for cases where $T$ of the forward diffusion process is large, as in the case of EDM (Karras et al., 2022), such a choice would amplify the model outputs which may cause poor performance. As such, we choose:

$$
c_{\text{out}}(t) = \frac{T - t}{T}
\tag{15}
$$

to ensure that the model is multiplied by a factor no larger than 1. For the skip connection function, $c_{\text{out}}(t)$, let's begin by examining the solution for the provided Probability Flow ODE system:

$$
\mathbf{x}_t = \mathbf{z} + \int_T^t \frac{\mathbf{x}_{t'} - D_\phi(\mathbf{x}_{t'}, t')}{t'}\, \mathrm{d}t'.
\tag{16}
$$

Given that the constant function, $c_{\text{skip}}(t) = 1$, already meets its boundary condition at $T$, it might appear to be the obvious choice for the skip function. However, when considering Equation 16 and Equation 14, it can be seen that the model, $\mathbf{X}_\theta$, when it solves the ODE system is such that:

$$
\begin{aligned}
\mathbf{X}_\theta(\mathbf{z}, t) = & \frac{1}{c_{\text{out}}(t)} \int_T^t \frac{\mathbf{x}_{t'} - D_\phi(\mathbf{x}_{t'}, t')}{t'}\, \mathrm{d}t' \\
= & \frac{T}{T - t} \int_T^t \frac{\mathbf{x}_{t'} - D_\phi(\mathbf{x}_{t'}, t')}{t'}\, \mathrm{d}t'.
\end{aligned}
\tag{17}
$$

At time $t = 0$,

$$
\begin{aligned}
\mathbf{X}_\theta(\mathbf{z}, 0) = & \int_T^0 \frac{\mathbf{x}_{t'} - D_\phi(\mathbf{x}_{t'}, t')}{t'}\, \mathrm{d}t' \\
= & \mathbf{x}_0 - \mathbf{z}.
\end{aligned}
\tag{18}
$$

Given the contrasting magnitudes of $\mathbf{x}_0$ within the range $[-1, 1]$ and $\mathbf{z}$ which has significantly larger values due to its high variance, aligning our model, $\mathbf{X}_\theta(\mathbf{z}, 0)$, with $\mathbf{x}_0$ at $t = 0$ emerges as a superior choice. More specifically:

$$\mathbf{X}_\theta(\mathbf{z}, 0) = \mathbf{x}_0$$
$$= \mathbf{z} + \int_T^0 \frac{\mathbf{x}_{t'} - D_\phi(\mathbf{x}_{t'}, t')}{t'} \, dt'. \tag{19}$$

Thus, for any arbitrary time, $t$, we desire our model to be expressed as:

$$\mathbf{X}_\theta(\mathbf{z}, t) = \mathbf{z} + \frac{T}{T - t} \int_T^t \frac{\mathbf{x}_{t'} - D_\phi(\mathbf{x}_{t'}, t')}{t'} \, dt'. \tag{20}$$

By plugging in the above formula into Equation 14, considering the choice of $c_{\text{out}}$ as well as the solution of the probability flow ODE in Equation 16, we obtain the given skip connection:

$$c_{\text{skip}}(t) = \frac{t}{T} \tag{21}$$

For the choice of functions, $c_{\text{in}}(t)$ and $c_{\text{noise}}(t)$, we opted to use the same functions utilized in the teacher EDM model. Additionally, the time value of the in function, $c_{\text{in}}(T)$, is set to $T$, given that the input consists of noise, representing the distribution corresponding to time $T$.

### A.5 Trajectory comparisons

In Figure 8, we present a comparison of the trajectories obtained from both the teacher EDM model and the student PID model. Notably, for the same noise seed, we observe that the trajectories represented by $x_\theta(\mathbf{z}, \cdot)$ in the student model align remarkably well with those of the teacher model, with only minor aberrations. This demonstration underscores the ability of our model to predict all points along the trajectory in a continuous manner and not only the origin.

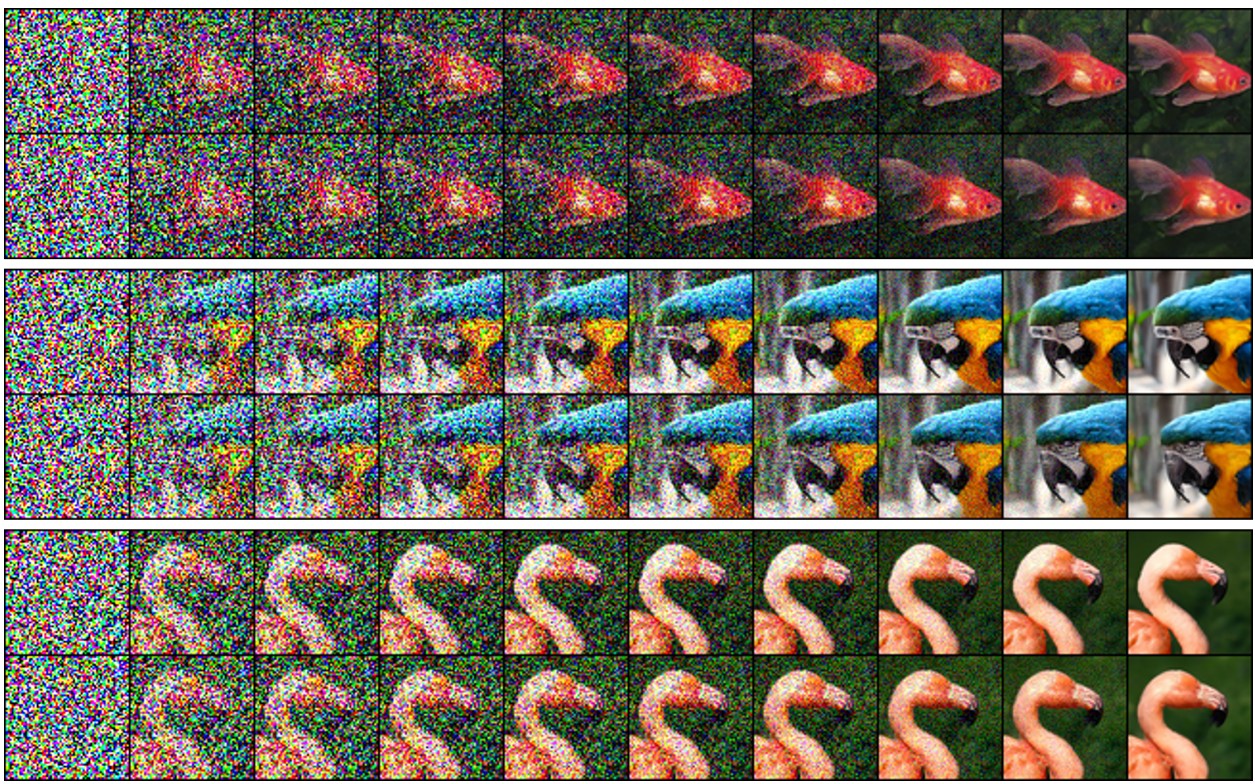

Figure 8: Trajectory comparisons on ImageNet with EDM teacher (top) and PID student (bottom).

## A.6 Additional Random Samples

In this section, we provide additional samples from our PID model for ImageNet 64x64 and CIFAR-10. The images are obtained by employing the same class for ImageNet 64x64 and applying the same noise seed to both the teacher EDM and student PID models.

### A.6.1 CIFAR-10

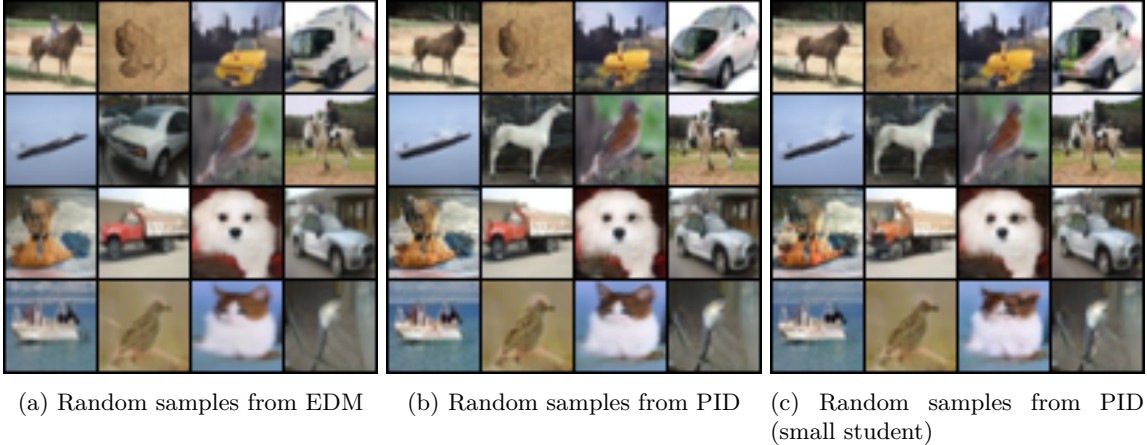

(a) Random samples from EDM     (b) Random samples from PID     (c) Random samples from PID (small student)

Figure 9: Unconditional image generation comparison on CIFAR-10 for the same seed. Left panel: random samples generated by EDM teacher model. Middle panel: generated by PID student model. Right panel: generated by small PID student model.

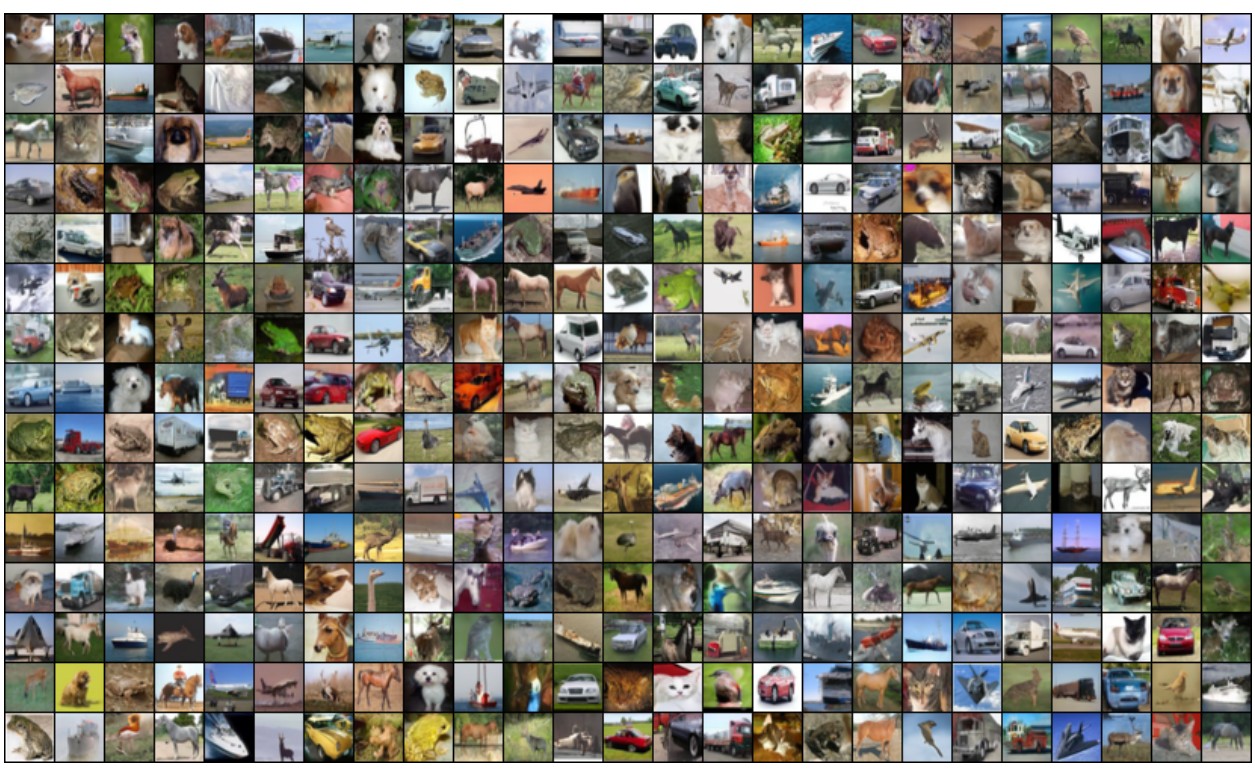

(a) Random samples from EDM (FID=2.04)

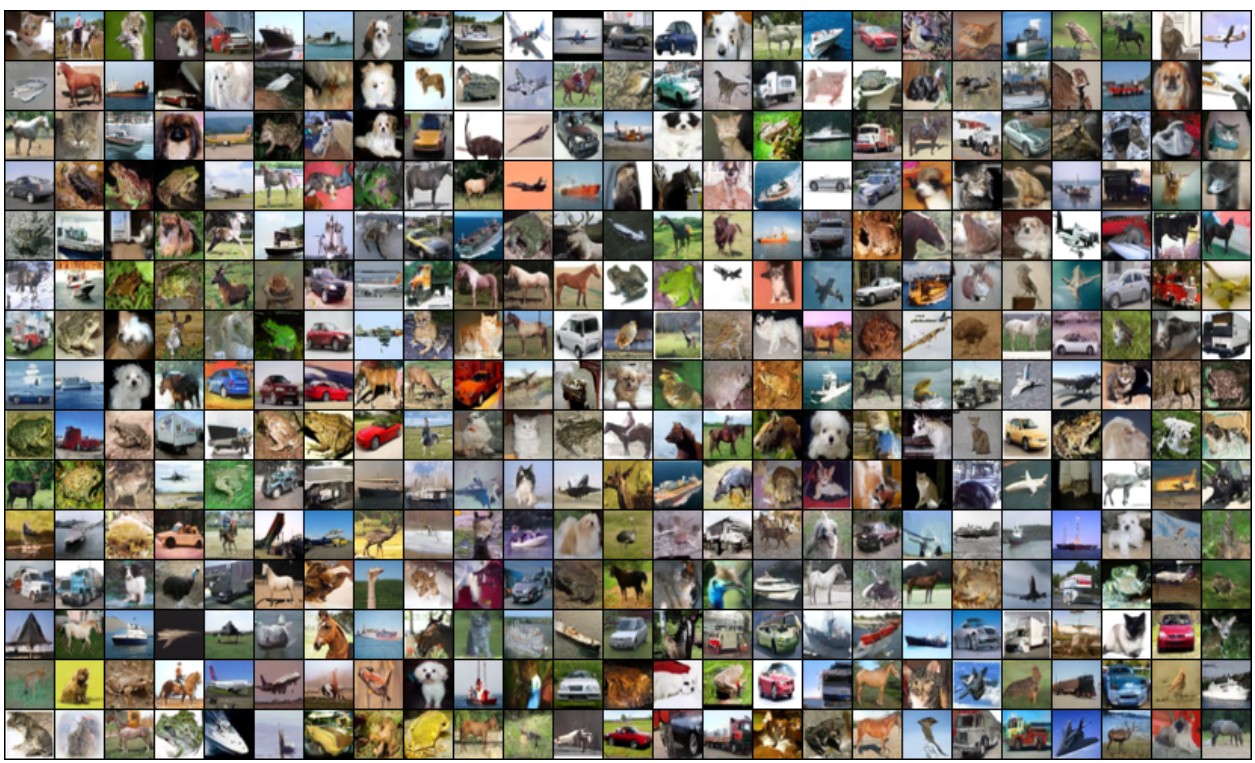

(b) Random samples from PID (FID=3.92)

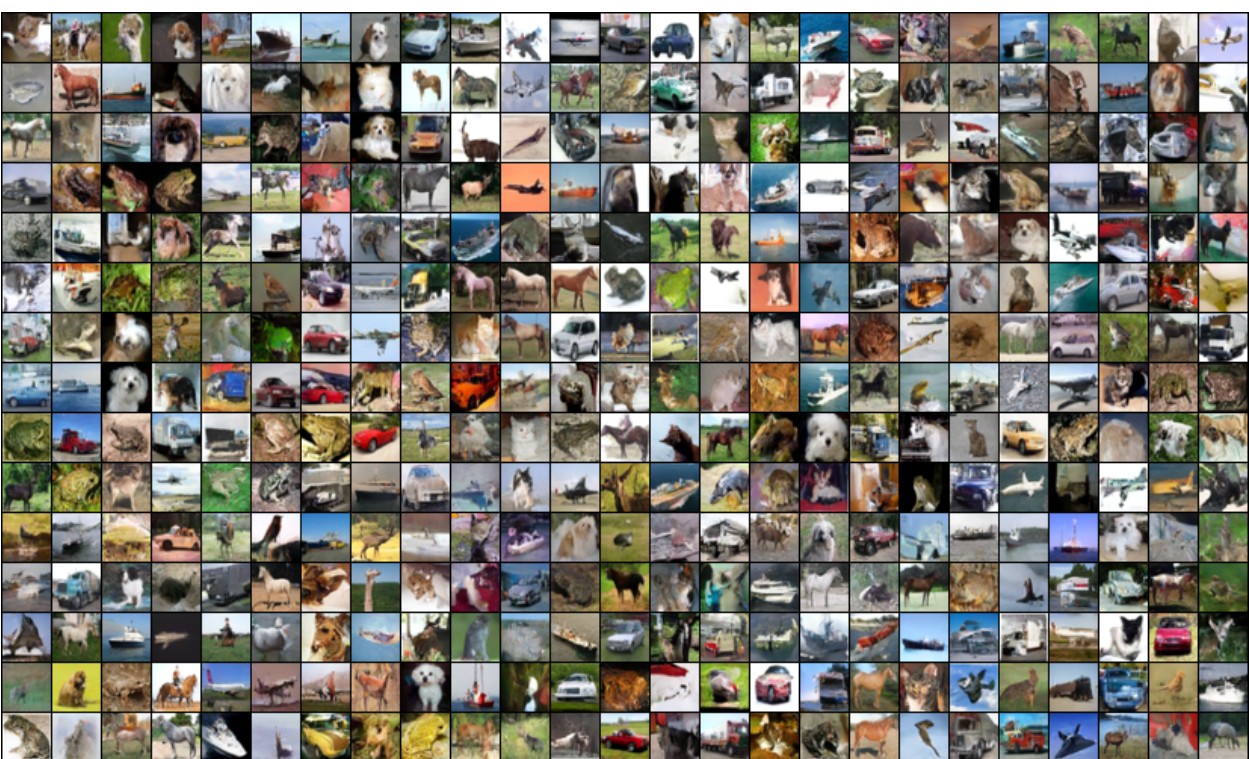

(c) Random samples from PID small student (FID=8.29)

Figure 10: Unconditional image generation comparison on CIFAR-10 for the same seed.

### A.6.2   ImageNet 64x64

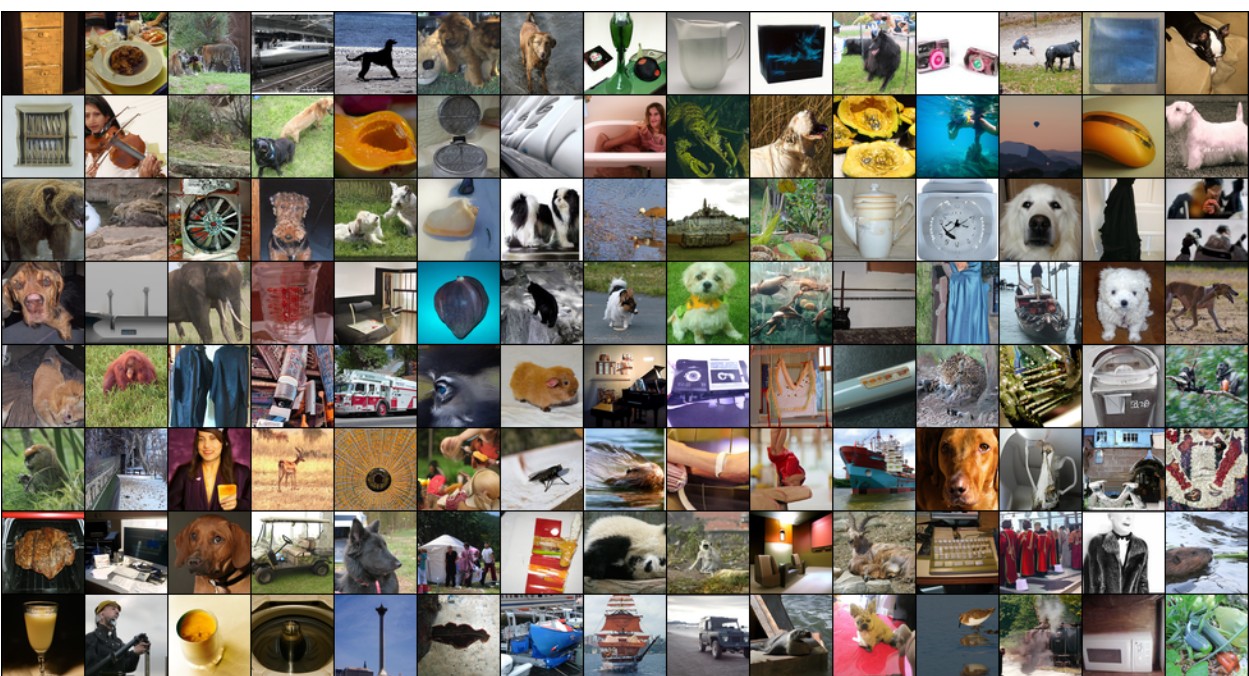

(a) Random samples from EDM on ImageNet (FID=2.44)

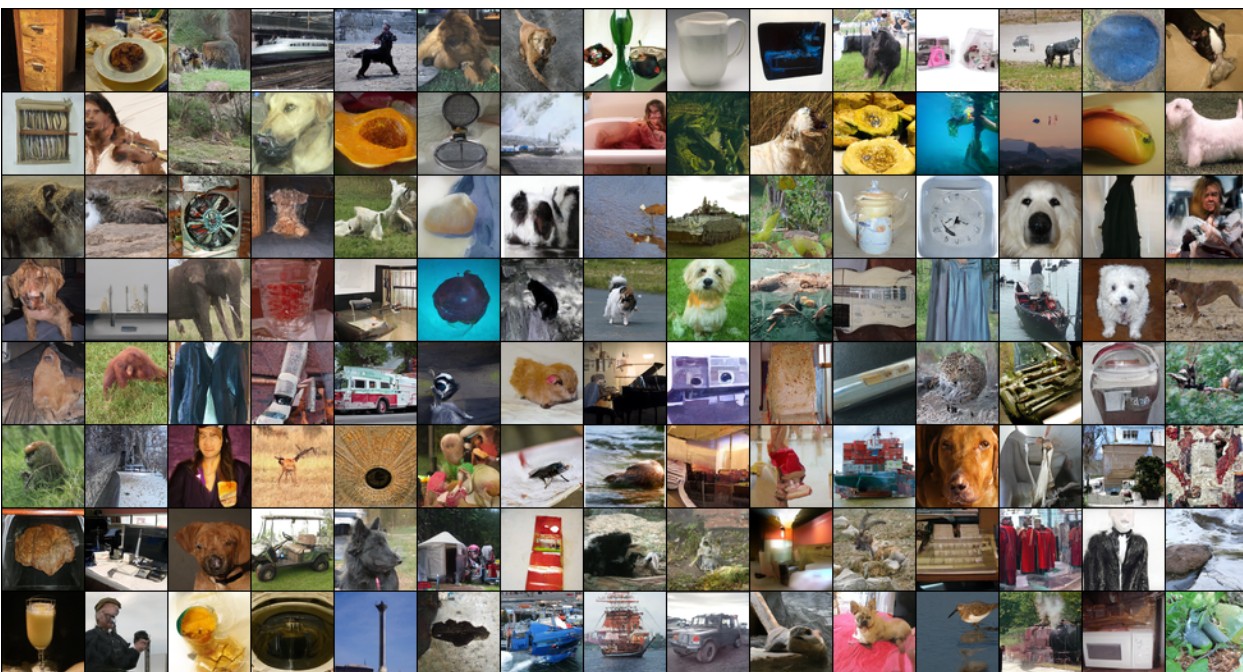

(b) Random samples from PID on ImageNet (FID=9.49)

Figure 11: Conditional image generation comparison on ImageNet 64×64 for the same seed.

