# OpenReview forum: "Physics Informed Distillation for Diffusion Models"
_TMLR — Accepted by TMLR_

### Review · Reviewer_hbhX · 2024-02-08

**Summary Of Contributions:**

The authors propose Physics Informed Distillation (PID), a novel approach for distilling a single-step image generator from pre-trained diffusion models. This method offers the distinct advantage of eliminating the need for training images or sampled clean images, consequently reducing training costs. Empirical experiments validate the performance of PID, demonstrating its superiority over existing baselines.

**Audience:**

Yes

**Broader Impact Concerns:**

No ethical concerns

**Claims And Evidence:**

Yes

**Requested Changes:**

See above. Would like to hear the authors' response.

**Strengths And Weaknesses:**

The proposed method exhibits strong similarities to CD with minor modifications. Both focus on constructing a feedforward ODE solver whose dynamics align with a pre-trained diffusion model. However, while CD relies on ground truth samples from p(x_t) and training data p(x_0), PID acquires samples from the student network itself. Intriguingly, PID outperforms CD.

I conjecture that the method's robustness might be linked to initialization—student network initialization heavily influences distillation success. In some ways, PID resembles forward KL divergence, where the optimized distribution (the student network) must generate samples for loss evaluation. Conversely, CD's behavior is closer to reverse KL with its sampling process. Improper student initialization could lead to local minima if initial samples significantly deviate from the teacher model's training distribution, where the teacher model lacks strong generalization.

I'd appreciate the authors' insights on these points. Additionally, while inspired by the EDM work, the claim of 'Physics Informed Network' seems somewhat exaggerated as the core contribution lies in the denoiser network parameterization.

---

> ### Author Response · Authors · 2024-02-23
> **Response to Reviewer hbhX**
>
> We thank the reviewer for raising interesting points for discussion. Our response is provided below.
>
> > **Random initialization:**
>
> Thank you very much for raising this point. The Table below shows the results from CIFAR-10 at 200k iterations. From this, we observed that as conjectured, initializations indeed do play an important role in obtaining good FID performance. We hypothesize that this issue could be mitigated through some form of discretization curriculum similar to that used in Consistency Training [1] but we leave that for future works. Discussion on this is included in Appendix A.3.2 of the revised manuscript.
>
> | Initialize from teacher | FID  |
> |-------------------------|------|
> | No                      | 7.52 |
> | Yes                     | 4.46 |
>
> > **Reason for the name 'Physics Informed':**
>
> The main reason for the name 'Physics Informed' comes from our attempt to pay homage to the field 'Physics Informed Neural Networks' in which a lot of our inspiration such as our distillation loss is drawn from which alot of our inspiration such as our distillation loss is drawn from.
>
>
>
>
> ---
> [1] Song, Yang, et al. "Consistency models." In International Conference on Learning Representations, 2022.

---

### Review · Reviewer_chFg · 2024-02-12

**Summary Of Contributions:**

This paper proposes to distill a (multistep by nature) diffusion model into a single-step network (PID). This network takes as input the initial noise and the diffusion time, and imitates the whole probability flow ODE trajectory. Taking inspiration in the literature of Physics-Informed Neural Networks (PINNs), this network is trained so that its numerical temporal first variation approaches the score estimated by the teacher model at any time. Inference is achieved by querying the network with an initial noise and the final diffusion time.

Several adjustments to the usual PINNs framework are proposed to alleviate issues arising from the probability flow ODE's particularities, in particular by changing the PINN loss and the parameterization of the student network. The generation performance and temporal efficiency of the proposed method are then assessed w.r.t. competitive methods and via ablation studies.

**Audience:**

Yes

**Claims And Evidence:**

No

**Requested Changes:**

I believe that most of the issues I described above can be addressed in a revision.

**Essential changes.**
- Improved empirical evidence supporting the claims of temporal efficiency.
- Completed discussion of related work.

**Desirable changes.**
- Improved reproducibility (code, more details in appendix, etc.).
- Empirical justification of numerical differentiation.
- Less focus on the straightforward theory.

**Quality changes.**
Cf. description above.

**Strengths And Weaknesses:**

## Strengths

PID tackles a **relevant and challenging problem**. Reducing the inference cost of diffusion models is an important line of work with direct practical implications. It also helps researchers design more cost-efficient models, as for consistency models.

The paper is overall **clear, well-written and easy to read**. PID benefits from **relevant design choices**: the adaptation of PINNs to this problem is natural, and supported by some well-motivated changes to the original setting. This results in a **simple method**, which is commendable as it makes it easily applicable.

The experimental results demonstrate the **competitiveness of the approach**, as precisely claimed in the paper. While PID is not state-of-the-art, I believe it can still be interesting for the community via its simplicity, potential applicability, and potential follow-ups.

## Weaknesses

### Supporting Claims on Temporal Efficiency

Besides the competitive results, one of the experiments' claim is that it has an advantageous training efficiency. This claim is **insufficiently supported**, for four reasons.
1. The paper insists that methods such as DSNO are expensive because they necessitate synthetic data. This should be quantified as the paper does not provide any element of comparison.
2. The only provided comparison deals with iteration speed (w.r.t. PD and CD). While this is useful information (provided that batch size is the same), this does not provide the full picture, i.e. full training time.
3. I do not understand why the model needs less hyperparameter tuning (claimed in Sections 1 and 6.4). The choice of discretization level has an impact on the performance, and could have an impact on the required training time.
4. In Section A.1, it is stated that all student models were initialized with the weights of the pretrained models. To ensure fair comparison, it should be clear that this is the case for all applicable models.

### Related Work

The paper does not, or fails to, contextualize its contribution w.r.t. to two closely related methods. This should be fixed so that the contributions of the paper are clarified.

The first one is **BOOT** (Gu et al., 2023). It is cited in the paper but only scarcely discussed. It is said to be similar to PID but on a different ODE. This requires further explanations: is the ODE choice the only difference between both methods? is there any specificity/advantage of PID?

The second one is **CTM** (Kim et al., 2024). I know that this paper and this submission are almost concurrent so I will not judge the novelty of the submission on this point, but I believe this deserves some discussion and possibly a comparison in the experiments. Indeed, to my understanding, although presented in a different fashion, CTM encompasses PID (same principle of learning the solution, possibility to rely on a single ODE discretization time step, similar loss) and also generalizes it to intermediate initializations and multi-step generation.

Less importantly, I would suggest the authors to include Jolicoeur-Martineau et al. (2021) and Karras et al. (2022) in the "Training-free methods" paragraph of their related work section, as they leveraged a second-order solver to reduce NFE of diffusion models.

Jolicoeur-Martineau et al. Gotta Go Fast When Generating Data with Score-Based Models. arXiv, 2021.\
Kim et al. Consistency Trajectory Models: Learning Probability Flow ODE Trajectory of Diffusion. ICLR 2024. [public since October 1st, 2023]

### Reproducibility

The paper's reproducibility could be improved.
- There is no source code and no promise to release it.
- Algorithm 1 does not specify the batching strategy: how are time indices samples in a batch (all the same or i.i.d.)?
- The appendix contains no specification of hardware or software.

### Justification of Numerical Differentiation

The use of numerical instead of automatic differentiation **should be motivated by empirical evidence**. I understand that Chiu et al. (2022) showed that automatic differentiation can lead to bad solutions for PINNs, but this concerns, to my understanding, the case of few sampling points. It remains to be shown that this problem transfers to the setting of the diffusion ODE, and an experimental result would show the advantage of the chosen approach.

### Straightforward Theory

I understand the motivation for a theory supporting the proposed approach, but in this case **it is not an important contribution** and does not deserve a full theorem.

In a nutshell, the result states that if PID perfectly imitates the Euler-approximated diffusion ODE (loss equals to 0), then the approximation error of PID is the same as Euler. This is straightforward and does not provide much insight. This result should be downgraded to a remark / trivial lemma supporting the soundness of the approach.

### Minor Issues

- I have reservations regarding the qualification of the method as "physics-informed". I understand the inspiration from PINNs, but there is no physics here.
- $D_{\phi}$ is not the teacher diffusion model, but the teacher denoiser network (p. 4).
- Notations in Section A.2 differ from the main paper (in particular $\mathbf{x_0}$).
- Quality issues:
  - the norm in Eq. (5) should be adjusted to the height of its argument;
  - the second mention of BOOT (in Section 4.1) should be referenced to improve text clarity;
  - "discritization" should be "discretization" (p. 9);
  - references should be checked to cite the most recent version papers (e.g. Consistency models which was published at ICML 2023);
  - in Section A.2, $x'\_t$ should be $x\_{t'}$.

---

> ### Author Response · Authors · 2024-02-23
> **Response to Reviewer chFg (Response 1/2)**
>
> We thank the reviewer for their thoughtful comments. Our responses to the reviewer’s specific concerns are as below:
>
> > **1. Supporting Claims on Temporal Efficiency:**
> - **Response to Comment 1 & 2:** The main gain in computational cost of PID over DSNO [1] comes from the need for such methods to construct a trajectory dataset prior to distillation training. This additional training phase substantially increases the computational overhead. For instance, in the ImageNet 64x64 experiment conducted with DSNO, a 16-step PD model [2] was employed to generate the trajectory dataset instead of utilizing a teacher diffusion model. Incorporating the time required to train a 16-step PD model, the additional cost to training would be approximate 1536 A100 GPU hours. Regrettably, since DSNO did not provide the total training iterations in their paper, we lack the necessary information to calculate the total training time for their experiments. However, we have included the total training time for CD [3], PD [2], and PID in Appendix A.3.4. In this section, it can be seen that PID is slower than CD, primarily due to the necessity of utilizing two functional evaluations for numerical differentiation.
> - **Response to Comment 3:** In Section 6.4, the need for less hyperparameter tuning was refering to the results in Section 6.2 and Theorem 1. Notably, our results showcase a predictable trend with regards to discretization number both theoretically and empirically, where increased discretization number improves performance up to saturation point. Additionally, a higher discretization number has no effect on training time as seen in Figure 3 in our main manuscript as it not only converges faster to a better FID value in the same number of iterations. This is in stark contrast to methods such as CD [3] where increasing or decreasing discretization numbers away from its optimal discretization number results in performance degradation resulting in the need to search this optimal discretization number that differs from dataset to dataset.
> - **Response to Comment 4:** In our experiments, all our (PID) student models were initialized with the Teacher EDM [4] weights. For the competing methods, all methods were also initialized with their respective teacher diffusion model weights. In Table 1 and 2, we specify the teacher models used as specified in their respective papers.
>
> > **2. Related Works:**
> - **Response to Comment 1:** For the differences with the recent work BOOT [5], the differences start from our focus on the original ODE and BOOT's focus on Signal ODEs. As our work focuses on solving the original probability flow ODE system, we tackle issues such as lipchitz explosion problems near the origin that cause instability in training via our parametrization as described in Appendix A.2 in our main manuscript. For BOOT, the focus on Signal ODE alleviates this issue but introduces a new problem, the inability to satisfy boundary conditions through hard constraints. As such, starting from the different ODE systems tackled by BOOT and our approach, our methods diverge due to the different difficulties present in both ODE systems. We have added a paragraph discussing this in Section 6 of the revised manuscript.
> - **Response to Comment 2:** While there are similarities in CTM [6] and PID, where we both distill the Probability Flow ODE system, notable differences are also present. It is worth emphasizing that CTM takes a similar view as CT, where they distill the paths traced by the numerical solvers. On the other hand, we view training from a PINN perspective, where the loss associated with the ODE system is minimized during training. Note that the similarities between CTM and PID exist only in the Euler method due to its relationship with first order gradient approximation. For second order numerical gradient approximations as in Table 3, our method works well despite differing from CTM training due to this difference in theoretical foundation of both approaches. As such, CTM [6] and PID can be viewed as methods arguing from different perspectives, being tangentially equivalent only in the first order setting. We have added a paragraph discussing this in Section 6.
> - **Response to Comment 3:** We thank the reviewer for pointing this out. Mention of the 2 papers have been added in the revised manuscript.

---

> ### Author Response · Authors · 2024-02-23
> **Response to Reviewer chFg (Response 2/2)**
>
> > **3. Reproducibility:**
> - **Response to Comment 1:** The code and model checkpoints will be released upon acceptance.
> - **Response to Comment 2:** The time indices are sampled i.i.d from a uniform distribution.
> - **Response to Comment 3:** The hardware used is specified in Appendix A.1 and the code was done in Pytorch.
>
> > **4. Justification of Numerical Differentiation:**
> - **Response to Comment 1:** We have added an additional ablation study on the importance of Numerial Differentiation in Appendix A.3.1. In Chiu et al. (2022) [7], Figure 11(a) highlights a persistent performance gap between Automatic Differentiation and Numerical Differentiation, even at high numbers of collocation points within a system of partial differential equations. It appears that instances where Automatic Differentiation appears to outperform Numerical Differentiation occur predominantly in single-variable ODE systems, which deviates significantly from our scenario. We speculate that a comparable situation to the one observed in Chiu et al. (2022) [7] in their system of partial differential equations may be contributing to the significantly worser performance of Automatic Differentiation.
>
>
> > **5. Minor Issues:**
> - **Response to Comment 1:** The main reason for the use of the term Physics-Informed is due to the fact that our loss is simply the PINN loss making our network a special type of PINNs.
> - **Response to Comment 2:** $D_{\phi}$ is the teacher diffusion model which can equivalently be viewed as the teacher denoiser model.
> - **Response to Typos and word choice:** We thank the reviewer for pointing out these improvements. We have adjusted them according to the suggestions in the revised manuscript.
>
> ---
> [1] Zheng, Hongkai, et al. "Fast sampling of diffusion models via operator learning." International Conference on Machine Learning. PMLR, 2023.
>
> [2] Salimans, Tim, and Jonathan Ho. "Progressive Distillation for Fast Sampling of Diffusion Models." International Conference on Learning Representations. 2021.
>
> [3] Song, Yang, et al. "Consistency models." In International Conference on Learning Representations, 2022.
>
> [4] Karras, Tero, et al. "Elucidating the design space of diffusion-based generative models." Advances in Neural Information Processing Systems 35 (2022): 26565-26577.
>
> [5] Gu, Jiatao, et al. "Boot: Data-free distillation of denoising diffusion models with bootstrapping." ICML 2023 Workshop on Structured Probabilistic Inference {\&} Generative Modeling. 2023.
>
> [6] Kim D, Lai C H, Liao W H, et al. Consistency Trajectory Models: Learning Probability Flow ODE Trajectory of Diffusion[C]//The Twelfth International Conference on Learning Representations. 2023.
>
> [7] Chiu, Pao-Hsiung, et al. "CAN-PINN: A fast physics-informed neural network based on coupled-automatic–numerical differentiation method." Computer Methods in Applied Mechanics and Engineering 395 (2022): 114909.

---

> ### Comment · Reviewer_chFg · 2024-02-25
> **Update by Reviewer chFg**
>
> I would like to thank the authors for their response, which I carefully read together with the paper's revision.
>
> ## Overall Comment
>
> The authors' response address most of my concerns. Two notable exceptions are the claims of efficiency w.r.t. methods necessitating synthetic data like DSNO, and the overestimated theoretical contribution.
>
> Nonetheless, I think that the provided explanations and changes could be better integrated into the paper. Some important details are only the authors' comments and should be included in the paper. Most of the changes to the paper, especially the additional ablation studies, were added in the appendix; a smoother integration in the different sections of the main paper would better highlight their importance in supporting its claims. Finally, I would have kept the related work section at the beginning of the paper; this would allow the reader to better appreciate the difference with BOOT in the experiments.
>
> Let me follow-up on the raised issues below.
>
> ## Supporting Claims on Temporal Efficiency
>
> 1. The provided elements of comparison are interesting. They should be included in the paper; I do not see them in the current revision. However, they are not sufficient to conclude, especially as PID takes $197 \times 64 = 12608$ A100 GPU hours (way above the overhead of DSNO). Given that the source codes of both DSNO and Rectified Flow are public, an estimation of their training time could be provided to enter Table 7 and support the claims of efficiency made in the introduction, related work and experiments.
> 2. The provided numbers on training time do address my comment.
> 3. The additional explanation is convincing. I think it deserves to be included in the paper.
> 4. The additional details are conclusive. They should be in the paper as well.
>
> ## Related Work
>
> Overall, the response and revision properly address my requests. I would simply ask the authors to amend their description of Karras et al. (2022) which does not solely focus on the diffusion SDE.
>
> ## Reproducibility
>
> 1. While providing the code during the review process would have been a better option, I am satisfied with the promise to release it, provided it is clean and easy to reproduce (version numbers, launch commands, etc.). Releasing model checkpoints is a nice addition.
> 2. Duly noted; the information could be included in the appendix.
> 3. I think that, besides the number and model of GPUs, the appendix should at least cite PyTorch accordingly (Paske et al., 2019).
>
> Paske et al. PyTorch: An Imperative Style, High-Performance Deep Learning Library. NeurIPS 2019.
>
> ## Justification of Numerical Differentiation
>
> I am convinced by the additional explanations and experiments. They probably deserve to be included in the main paper (in both Sections 3 and 4).
>
> ## Straightforward Theory
>
> The authors did not address my comment on the theory. I maintain my original recommendation.
>
> ## A Few Other Minor Issues
>
> - "lipschitz" in Section 6 (p. 10) should be capitalized.
> - References to Karras et al. (2020) and Jolicoeur-Martineau et al. (2021) should be inlined.
> - The reference to DSNO should target its most recent conference acceptance (ICML 2023).
> - There is a formatting / incomplete sentence issue in Section A.3.1.

---

> ### Author Response · Authors · 2024-02-29
> **Response to Reviewer chFg**
>
> Thanks for the thoughtful reply.
>
> - **DSNO full time comparison:** Concerning the time comparisons for DSNO, we've identified a bug in the official code relating to gradient accumulation and total iteration. This bug results in the total iteration being contingent upon the number of gradient accumulation steps, consequently leading to fewer actual training iterations than specified in the configurations file. Due to these issues, we are not confident in comparing total training time, as training iterations aren't clearly defined as a result of this bug. Furthermore, in terms of iterations per second, DSNO lags behind PID due to its requirement of four functional evaluations per batch element, in contrast to PID's two.
>
> - **Straightforward Theory:** We have changed the theorem to a lemma and moved the proof of it to the appendix in accordance with the reviewers suggestion.
>
> - **Other changes:**  In the revised version, as suggested by the reviewer, we have moved the additional ablation studies to the main paper. Additionally, we have also fixed the typos and added clarifications on experimental settings as pointed out by the reviewer. We thank the reviewer again for pointing out these mistakes.

---

### Review · Reviewer_vn5G · 2024-02-15

**Summary Of Contributions:**

This paper introduces a Physics-Informed method for distilling a pretrained teacher diffusion model into a student diffusion model that can generate samples in a single step while maintaining sample quality that is close to the teacher model. The method is based on a synergy between Physics-Informed Neural Networks (PINN) and the ODE formulation that describes the sampling paths of diffusion models. Since PINNs are naturally suited for learning the evolution of arbitrary ODEs, the work observes that one can learn the entire diffusion ODE given an initial condition during training time and try to guess the final ODE state given an initial noise sample during test-time, thereby yielding single-step generation. Several practical considerations (residual formulation to deal with initial conditions, numeric 1st order gradients, re-arranged loss with stop gradient added to one term) are introduced. Experimental results show the method is competitive with distillation methods like Consistency Models, without requiring any samples from the teacher diffusion model.

**Audience:**

Yes

**Broader Impact Concerns:**

Broader impacts are not discussed, although this does not affect my evaluation of the work.

**Claims And Evidence:**

Yes

**Requested Changes:**

I only have minor change requests at the moment. I suggest that the authors annotate which methods use LPIPS and which do not when presenting their experimental results. I also would appreciate more discussion/interpretation of the use of stop-gradient.

**Strengths And Weaknesses:**

**Strengths**
* The primary strength of this work is the straightforwardness and naturalness of the proposed method. The authors observe a clear connection between PINN and diffusion distillation and show that standard ideas from each domain can be combined into an effective distillation method. I found this connection interesting and believe it could inspire future works in a similar direction.
* The paper was well-written and easy to understand. It provided suitable context for readers without much background in PINNs (such as myself).
* The experimental results demonstrate that the method can achieve reasonably strong single-step distillation results compared to relevant prior works.

**Weaknesses**
* The recent work BOOT explores similar ideas, as discussed by the authors. However, in my opinion, the formulation presented in the present work is somewhat clearer.
* The method relies on LPIPS to provide a significant boost to image quality. In this case, the method is not entirely self-contained because it implicitly relies on the perceptual dataset used to train LPIPS. While this design choice is shared by provide works, it does complicate comparisons with other works. I suggest notating which distillation works use LPIPS and which do not use LPIPS when presented experimental results such as Table 1 and 2.
* Several of the practical considerations complicate the otherwise clear method. In particular, the numeric 1st order gradients could be a bottleneck that prevent the method from achieve higher quality results. The use of stop-gradient seems somewhat arbitrary.

---

> ### Author Response · Authors · 2024-02-23
> **Response to Reviewer vn5G**
>
> We appreciate the reviewer's thorough review and constructive feedback, which will aid in enhancing the manuscript. Below, we address the requested changes.
>
> > **Notating the usage of LPIPS in the experiment results:**
>
> Thanks for the suggestion. In the revised version, we annotate the usage of the distance metric used in the respective distillation methods.
>
> > **Discussion on the usage of the stop-gradient:**
>
> The FID of a model trained on CIFAR-10 dataset without stop-gradient at 200k iterations is shown below. From this, we observe a drop in performance when the stop gradient was removed. We hypothesize that the removal of the stop gradient which updates the student while backpropagation through the teacher diffusion model behaves akin to an adversarial attack on the teacher model. This results in generated samples with low residual PINN loss but high levels of distortion in the images. We have added this discussion in Appendix A.3.3 in the revised manuscript.
>
> | Stop-Gradient | FID  |
> |---------------|------|
> | No            | 9.93 |
> | Yes           | 4.46 |

---

### Review · Reviewer_9HBp · 2024-02-21

**Summary Of Contributions:**

**Summary of the work**

Diffusion models are powerful ways of sampling from complex data distributions and provide superior performance over GANs and VAEs. However, since they require an iterative process to produce clean images from initial noise samples, their sampling speed can be slow.

This paper addresses this problem by using a distillation approach inspired from Physics Inspired Neural Networks (PINNs). The backward process of a diffusion model can be expressed in the form of a probability flow ODE. Instead of using an iterative solution, the paper uses a PINN to learn the solution trajectory of the ODE thereby resulting in a single step solution.

The effectiveness of their approach in generating image samples is shown with experiments on CIFAR-10 and ImageNet-64x64 datasets.

**Audience:**

Yes

**Claims And Evidence:**

Yes

**Requested Changes:**

Please see above.

**Strengths And Weaknesses:**

**Strengths**

- The key idea of the paper to use PINNs to model the ODE solution trajectories is well motivated and very naturally suited to diffusion models.

- Experimental results on CIFAR-10 and ImageNet-64x64 show that high quality image samples can be generated with the proposed physics informed distillation.
-
The paper is well written and easy to follow.

**Weaknesses**

I do not have any major comments on the paper’s weaknesses. A few minor comments are provided below.

- I am not entirely sure why the paper uses numerical differentiation over exact gradients that can be obtained using autodiff. The paper cites Chiu et al. to suggest that automatic differentiation can result in unphysical solutions. However, this appears to be the case only when there is a lack of enough grid points. Is that scenario relevant for the experimental setup used in the paper? Could the authors please clarify this?

- The authors mention a distillation approach, BOOT, in the related work that is similar to their approach but do not explain on how it differs from the proposed work. A better discussion on this would help better place this paper in the context of literature.

---

> ### Author Response · Authors · 2024-02-23
> **Response to Reviewer 9HBp**
>
> We appreciate the reviewer's positive feedback on our paper and address the concern raised below.
>
> > **The usage of numerical differentiation over exact gradients:**
>
> We have added an additional section in Appendix A.3.1, discussing this ablation study. Notably, when Automatic Differentiation is used, we found that the model converges to unrealistic images with high color contrast despite smaller ODE loss. Note that in Chiu et al. (2022) [1] Figure 11(a), it can be observed that in a system of partial differential equations, the performance gap between Automatic Differentiation and Numerical Differentiation still persists even at a high number of collocation points. To the best of my understanding, the times when Automatic Differentiation seems to outperform Numerical with sufficient collocation points is in the case of single variable ODE systems which is far from our case. As such, while this issue is more prevalent with a lower number of collocation points, even with a sufficiently large number of collocation points, this problem still seems to be present. We hypothesize that a similar situation to that observed by Chiu et al. (2022) [1] is the cause.
>
> |                           | FID   |
> |---------------------------|-------|
> | Automatic Differentiation | 90.60 |
> | Numerical                 | 7.03  |
>
> > **Discussion on revelant work BOOT [2]:**
>
> We have added a paragraph in the revised manuscript section 6 discussing it in more detail.
>
> ---
> [1] Chiu, Pao-Hsiung, et al. "CAN-PINN: A fast physics-informed neural network based on coupled-automatic–numerical differentiation method." Computer Methods in Applied Mechanics and Engineering 395 (2022): 114909.
>
> [2] Gu, Jiatao, et al. "Boot: Data-free distillation of denoising diffusion models with bootstrapping." ICML 2023 Workshop on Structured Probabilistic Inference {\&} Generative Modeling. 2023.

---

### Author Response · Authors · 2024-02-23
**Big thanks for taking the time to review our work!**

We're grateful to the reviewers for their valuable feedback. We've taken their suggestions seriously and have made updates to our paper, as outlined in our responses below. The revised manuscript is now uploaded, with tweaks to the tables and extra ablation experiments included in the appendix. We've highlighted the changes in red to make them easier to spot.

---

### Decision · Action_Editor_2LLk · 2024-05-05

**Recommendation:** Accept with minor revision

**Comment:**

The paper is strong with good numerical and theoretical evidence of the validity of the methods. The authors have answered most of the concerns of the reviewers. While it is not a state-of-the-art method in terms of distillation, this paper suggests that further experimental improvements could bridge the gap between the physics inspired PINN methods and consistency models. However, summarizing some of the points broudhgt by Reviewer chFg we would like the authors to update their manuscript on the following points.

* The details of training time comparison with DSNO should be included in the paper: GPU hours, training iterations time, bug in the public code.

* The remark on the reliance of CD in discretization parameters and the initialization of baselines should be in the paper (cf. Sections 1.3 and 1.4 of this response).

* A clean and easy to reproduce code as well as model checkpoints must be included in the final submission following the authors' promise.

* Following the downgrade of the theoretical part to a simple lemma, the theoretical contribution in the introduction (2nd item p. 2) should be reformulated and integrated into the first item.

* The choice of LPIPS instead of a distance metric in Section 4.3 should follow the lemma. It should be stated explicitly that it is an implementation choice, as the lemma does not cover LPIPS (which is not a distance metric contrary to the statement in Section 4.1 - to be corrected).

* All these changes should be smoothly integrated into the flow of the paper. Most of the changes to the paper, especially the additional ablation studies, were simply appended to the paper, whereas a smoother integration in the different sections would improve the writing. In particular, additional ablation studies should not be subsubsections of Section 6.5, and some of them might be better put together with other subsections (Section 6.5.1 with Section 6.1 or 6.2, Section 6.5.4 with Section 6.4).

**Audience:**

Yes. Distillation of diffusion models is an important topic in the generative modeling community.

**Claims And Evidence:**

The claims of the paper are supported by both experimental and theoretical results. The authors compare their approach with other state-of-the-art distillation methods for diffusion models. Evaluations are run on CIFAR and Imagenet 64x64.